# Apoptotic stress-induced FGF signalling promotes non-cell autonomous resistance to cell death

Florian J. Bock [1,2,3 ✉], Egor Sedov[4], Elle Koren[4], Anna L. Koessinger[1,2], Catherine Cloix[1,2], Désirée Zerbst[1,2], Dimitris Athineos [1], Jayanthi Anand[1], Kirsteen J. Campbell [1,2], Karen Blyth [1,2], Yaron Fuchs[4] & Stephen W. G. Tait [1,2 ✉]

Damaged or superfluous cells are typically eliminated by apoptosis. Although apoptosis is a cell-autonomous process, apoptotic cells communicate with their environment in different ways. Here we describe a mechanism whereby cells under apoptotic stress can promote survival of neighbouring cells. We find that upon apoptotic stress, cells release the growth factor FGF2, leading to MEK-ERK-dependent transcriptional upregulation of pro-survival BCL-2 proteins in a non-cell autonomous manner. This transient upregulation of pro-survival BCL-2 proteins protects neighbouring cells from apoptosis. Accordingly, we find in certain cancer types a correlation between FGF-signalling, BCL-2 expression and worse prognosis. In vivo, upregulation of MCL-1 occurs in an FGF-dependent manner during skin repair, which regulates healing dynamics. Importantly, either co-treatment with FGF-receptor inhibitors or removal of apoptotic stress restores apoptotic sensitivity to cytotoxic therapy and delays wound healing. These data reveal a pathway by which cells under apoptotic stress can increase resistance to cell death in surrounding cells. Beyond mediating cytotoxic drug resistance, this process also provides a potential link between tissue damage and repair.

[1] Cancer Research UK Beatson Institute, Garscube Estate, Switchback Road, Glasgow G61 1BD, UK. [2] Institute of Cancer Sciences, University of Glasgow, Garscube Estate, Switchback Road, Glasgow G61 1QH, UK. [3] Department of Radiotherapy (MAASTRO), GROW-School for Oncology and Developmental Biology, Maastricht University, 6229 ER Maastricht, The Netherlands. [4] Laboratory of Stem Cell Biology and Regenerative Medicine, Department of Biology, Technion Israel Institute of Technology, Haifa 3200003, Israel. ✉email: f.bock@maastrichtuniversity.nl; stephen.tait@glasgow.ac.uk

The cellular decision to live or die is fundamentally important in biology. Inappropriate cell survival has been causally linked to various diseases including cancer and autoimmunity[1]. In cancer, many therapies act by engaging apoptosis, and the degree of apoptotic sensitivity or apoptotic priming often correlates with therapeutic efficacy[1,2]. Therefore, understanding how cancer cells survive therapy should provide new ways to circumvent this and improve tumour cell elimination.

Mitochondrial apoptosis represents a major form of regulated cell death[3]. During apoptosis, mitochondria are permeabilised through a process called mitochondrial outer membrane permeabilisation or MOMP. Widespread MOMP effectively acts as cellular death sentence by releasing mitochondrial proteins, such as cytochrome c, that activate caspase proteases leading to rapid apoptosis[3]. Even in the absence of caspases, MOMP typically commits a cell to death, and is thus considered a point-of-no-return. Consequently, mitochondrial outer membrane integrity is tightly regulated by pro- and anti-apoptotic BCL-2 family proteins. Anti-apoptotic BCL-2 proteins prevent apoptosis by binding pro-apoptotic BAX, BAK and BH3-only proteins. During apoptosis, BH3-only proteins activate BAX and BAK, which subsequently promote MOMP. This process is exploited by BH3-mimetics, a new class of anti-cancer drugs[1]. By binding anti-apoptotic BCL-2 proteins, BH3-mimetics antagonise BCL-2 pro-survival function, sensitising cells to apoptosis[4]. Various BH3-mimetics have been developed that target select or multiple anti-apoptotic BCL-2 family members. Amongst them, the BCL-2 specific BH3-mimetic venetoclax[5] shows considerable clinical promise and is approved for the treatment of chronic lymphocytic leukaemia (CLL)[6] and in combination therapy to treat acute myeloid leukaemia (AML)[7,8]. However, in solid tumours, BH3-mimetics are typically less effective, implying that additional survival mechanisms must be targeted in order to maximise their potential.

We set out to identify mechanisms of apoptotic resistance using BH3-mimetics as tool compounds. Selecting for cells surviving venetoclax treatment, we found that resistance was associated with increased anti-apoptotic BCL-2 and MCL-1 expression. Surprisingly, resistance occurred in a non-cell autonomous manner. We find that under apoptotic stress, cells can release FGF2. In turn, FGF2 triggers MEK-ERK signalling, resulting in increased anti-apoptotic BCL-2 and MCL-1 protein expression and apoptotic resistance. In certain cancer types, we found a correlation between FGF-signalling, BCL-2 and MCL-1 expression and poorer patient prognosis. Furthermore, we find FGF-dependent signalling results in upregulation of MCL-1 during wound healing and promotes tissue repair. Together, these findings unveil a non-cell autonomous mechanism of apoptotic resistance, where apoptotic stress—via FGF signalling—promotes cell survival. As we discuss, this process may have wide-ranging roles in health and disease.

## Results

### BH3-mimetics and BH3-only proteins upregulate BCL-2 and MCL-1 causing apoptotic resistance.
We initially sought to define mechanisms of cell death resistance using BCL-2 targeting BH3-mimetics. For this purpose, we used our recently developed method called mito-priming[9]. In this system, cells co-express a pro-apoptotic BH3-only protein and an anti-apoptotic BCL-2 family member at equimolar levels and are therefore highly sensitive to BCL-2 targeting BH3-mimetic drugs (Fig. 1a). HeLa cells were used that stably express tBID together with BCL-2 (HeLa tBID-2A-BCL-2, hereafter called HeLa tBID2A). Cell viability was determined using livecell imaging following venetoclax

treatment using Syto21 to label all cells and propidium iodide to label dead cells. As expected, the majority of cells died rapidly following venetoclax treatment, nevertheless some cells failed to die (Fig. 1b). To investigate the mechanisms of venetoclax resistance, HeLa tBID2A cells were cultured continuously in venetoclax to select for resistant cells. Increased expression of pro-survival BCL-2 family proteins is a common means of apoptotic resistance[10]. Indeed, cells that were continuously or intermittently cultured in venetoclax displayed higher expression of anti-apoptotic BCL-2 and MCL-1 (Supplementary Fig. 1a). Surprisingly, following culture in regular medium post-venetoclax treatment, the resistant cells became sensitive to venetoclax again over time (Supplementary Fig. 1b). This decrease of resistance was accompanied by a decrease of BCL-2 and MCL-1 expression back to basal levels (Supplementary Fig. 1c). We next investigated whether short-term treatment with venetoclax was sufficient to promote BCL-2 and MCL-1 upregulation. Indeed, 3 h of venetoclax treatment led to increased levels of BCL-2 and MCL-1 (Fig. 1c). As before, BCL-2 and MCL-1 levels decreased following removal of venetoclax, demonstrating a reversible upregulation (Fig. 1d). This effect was not restricted to venetoclax, because an increase in BCL-2 and MCL-1 expression was also observed following treatment with other BH3mimetics (navitoclax and ABT-737) (Supplementary Fig. 1d). Given that venetoclax-induced upregulation of BCL-2 and MCL-1 is reversible, this suggests that it is not genetically based. We noted an initial resistance of HeLa tBID2A cells cultured in venetoclax to re-treatment with venetoclax (Supplementary Fig. 1b), presumably due to increased levels of BCL-2 and MCL-1. Since increased levels of BCL-2 and MCL-1 were also observed in response to acute treatment, we investigated whether this was also sufficient to protect from apoptosis. Indeed, treatment with venetoclax for 48 h could protect from re-treatment with venetoclax and S63845, a specific inhibitor of MCL-1[11] (Fig. 1e). This protection was dependent on the increased levels of BCL-2 and MCL-1, because increasing the dose of venetoclax and S63845 could overcome the resistance (Fig. 1e). The pro-apoptotic proteins BAX and BAK are essential for mitochondrial outer membrane permeabilization (MOMP) during apoptosis[12]. To determine the role of apoptosis in the upregulation of BCL-2 and MCL-1, we generated HeLa tBID2A cells deficient in BAX and BAK using CRISPR-Cas9 genome editing. As expected, BAX BAK-deleted HeLa tBID2A cells were completely protected from mitochondrial apoptosis and caspase activation in response to venetoclax treatment (Supplementary Fig. 1e, f). Nevertheless, despite an inability to undergo apoptosis, BAX BAK-deleted HeLa tBID2A cells still upregulated BCL-2 and MCL-1 following venetoclax treatment (Fig. 1f, Supplementary Fig. 1g). Similarly, upregulation of BCL-2 and MCL-1 was also observed when caspase activity was blocked using the caspase inhibitor qVD-OPh (Fig. 1f). These data demonstrate that while the upregulation of BCL-2 and MCL-1 requires BH3-mimetic treatment, it occurs irrespective of apoptosis. Finally, to determine whether upregulation of BCL-2 and MCL-1 was specific to BH3-mimetics, we examined if a comparable effect could be observed by expressing BH3-only proteins. Control or BAX and BAK deleted HeLa cells were transfected with BH3-only proteins (tBID, PUMA, tBID (BIM BH3, with the BID BH3 domain replaced with the BIM BH3 domain)) and analysed for MCL-1 expression by western blot (Fig. 1g, Supplementary Fig. 1h). In all cases, MCL-1 expression was upregulated, indicating that BH3-only proteins can have similar effects as BH3-mimetics. Collectively, these data demonstrate that BH3-mimetics and BH3-only proteins can promote apoptotic resistance by increasing pro-survival BCL-2 protein expression.

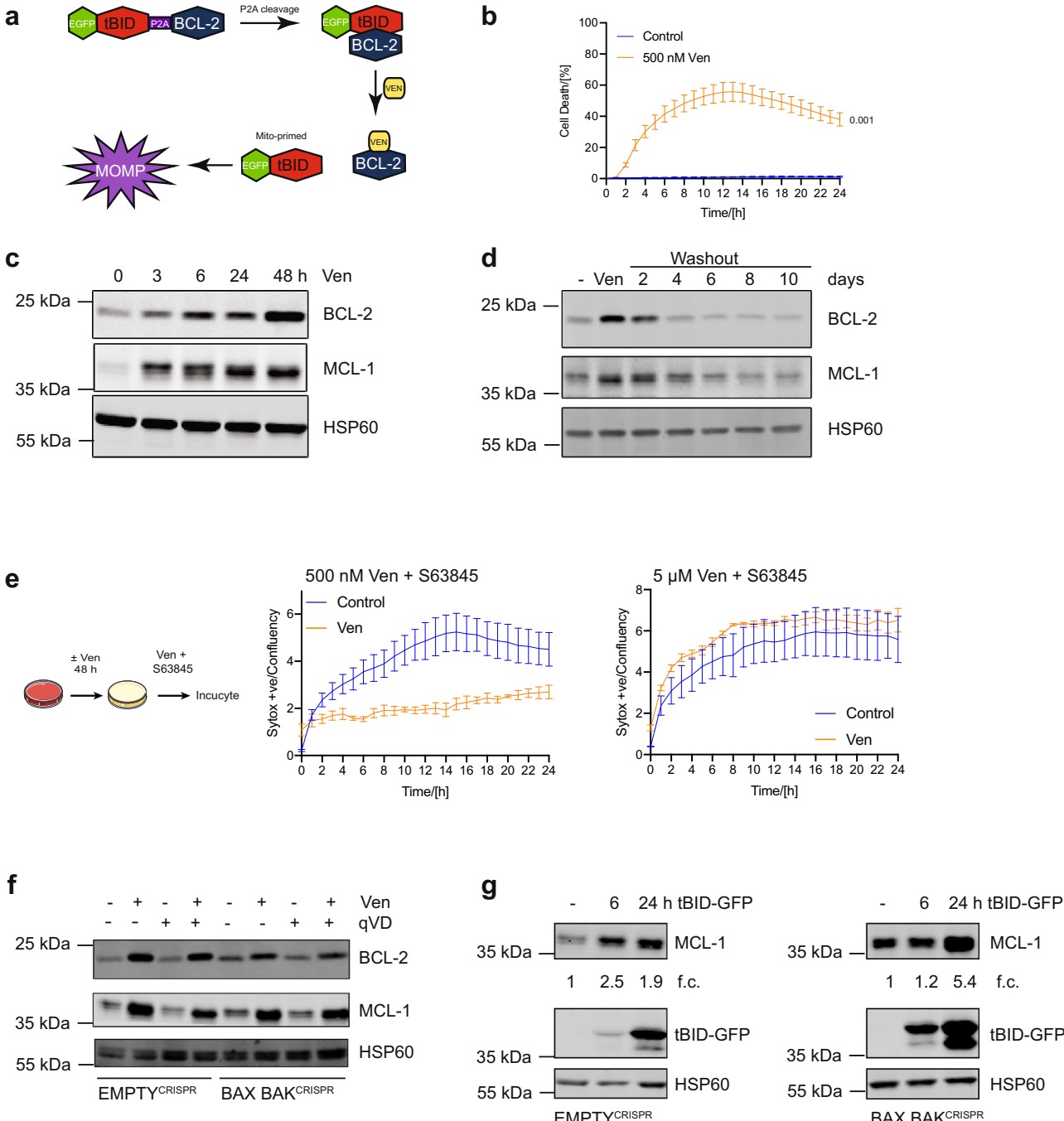

**Fig. 1 BH3-mimetics and BH3-only proteins upregulate BCL-2 and MCL-1 causing apoptotic resistance. a** Schematic model of the mitoprimed system. **b** HeLa tBID2A cells were treated with venetoclax (ven) and imaged over time. Percentage of dead cells was determined by staining all cells with Syto 21 and dead cells with propidium iodide. $n = 3$ independent experiments; mean values ± s.e.m.; unpaired, two-sided $t$-test at 24 h. **c** HeLa tBID2A cells were treated with 500 nM venetoclax, harvested at the indicated time points and protein expression was analysed by western blot (representative blot of three independent repeats). **d** HeLa tBID2A cells were treated for 48 h with 500 nM venetoclax followed by replacement with regular growth medium (washout). At the indicated times post medium change cells were harvested and protein expression was analysed by western blot (representative blot of three independent repeats). **e** HeLa tBID2A cells were treated with or without venetoclax as indicated for 48 h followed by treatment with venetoclax and the MCL-1 inhibitor S63845. Cell death was then monitored by Sytox Green staining and Incucyte imaging. $n = 3$ independent experiments; mean values ± s.e.m. **f** Control or BAX BAK[CRISPR] HeLa tBID2A cells were treated with 500 nM venetoclax in combination with 10 μM qVD-OPh as indicated for 48 h, harvested and protein expression was analysed by western blot (representative blot of three independent repeats). **g** Control or BAX BAK[CRISPR] HeLa cells were transfected with tBID-GFP plasmid, harvested after the indicated times and protein expression was analysed by western blot. Fold change normalised to loading control is stated below (representative blot of three independent repeats).

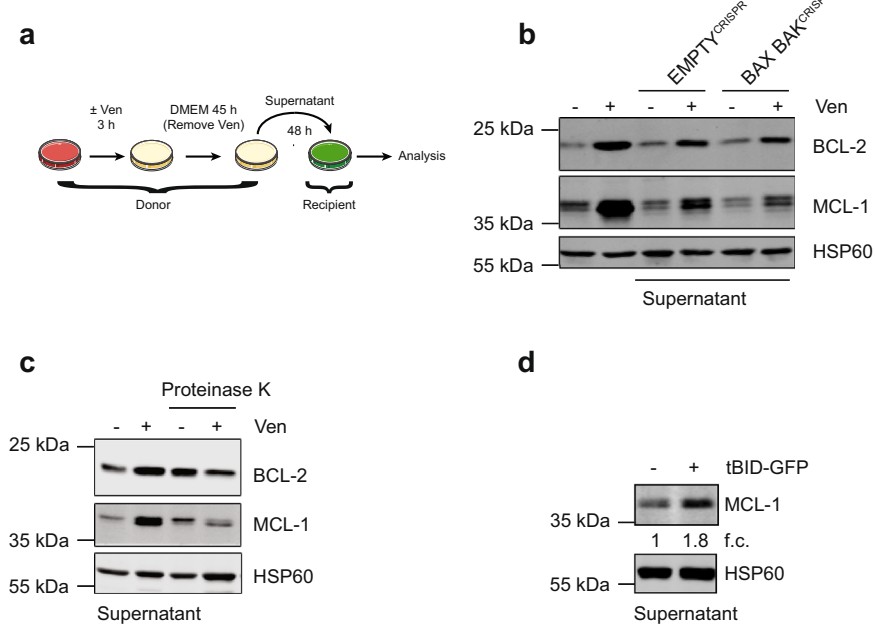

**Fig. 2 BH3-mimetics and BH3-only proteins can upregulate anti-apoptotic BCL-2 proteins in a non-cell autonomous manner. a** Schematic of supernatant transfer experiments: HeLa tBID2A cells were treated with 500 nM venetoclax for 3 h, followed by exchange to regular growth medium for 45 h. Supernatant was harvested, filtered and added onto recipient cells before analysis. **b** HeLa tBID2A cells were treated directly with venetoclax or with supernatant from untreated or venetoclax treated control or BAX BAK[CRISPR] cells as described in **a**. After 48 h cells were harvested and protein expression was analysed by western blot (representative blot of three independent repeats). **c** Supernatant from control or venetoclax treated cells was digested with Proteinase K before addition onto recipient cells and protein expression was analysed by western blot after 48 h (representative blot of three independent repeats). **d** Supernatant from HeLa cells transfected with tBID-GFP was collected after 48 h and transferred onto recipient HeLa cells. After 48 h, recipient cells were harvested and protein expression analysed by western blot. Fold change normalised to loading control is stated below (representative blot of two independent repeats).

**BH3-mimetics and BH3-only proteins can upregulate anti-apoptotic BCL-2 proteins in a non-cell autonomous manner.** To understand how venetoclax treatment causes upregulation of BCL-2 and MCL-1, we determined whether increases in protein or mRNA stability might contribute. HeLa tBID2A cells were treated with venetoclax for 24 h, after which inhibitors of protein synthesis (cycloheximide) or transcription (actinomycin D) were added for varying times. Neither MCL-1 nor BCL-2 protein or mRNA stability was increased after venetoclax treatment (Supplementary Fig. 2a, b), indicating that mechanisms besides protein or mRNA stability are likely responsible. Given these results, we investigated if venetoclax might upregulate BCL-2 and MCL-1 in a non-cell autonomous manner. Control or BAX BAK deleted HeLa tBID2A cells were treated for 3 h with 500 nM venetoclax, followed by exchange to regular medium for 45 h. Media from treated cells was then transferred to recipient cells, which were examined for BCL-2 and MCL-1 expression after 48 h (Fig. 2a). Importantly, media from venetoclax treated HeLa tBID2A cells promoted upregulation of BCL-2 and MCL-1 in recipient cells (Fig. 2b). Similarly, supernatant from BAX and BAK deficient cells also promoted MCL1 and BCL-2 upregulation, supporting earlier findings that cell death is not required for this effect (Fig. 2b). Supernatant from venetoclax treated cells failed to induce apoptosis in recipient cells, demonstrating the absence of potentially residual venetoclax (Supplementary Fig. 2c). Additionally, media from HeLa tBID2A cells treated with a different BCL-2 inhibitor, S55746[13], also upregulated MCL-1 and BCL-2 in recipient cells (Supplementary Fig. 2d). To investigate the mechanism of this non-cell autonomous effect, we first characterised the signal causing upregulation of anti-apoptotic BCL-2 proteins. Supernatant from venetoclax treated HeLa tBID2A cells was subjected to centrifugal filtration using a filter with a 3 kDa

cut-off. Flow-through and concentrate were added to recipient cells for 48 h, and MCL-1 and BCL-2 expression was determined by western blot (Supplementary Fig. 2e). Only the concentrate (containing molecules above 3 kDa) was capable of increasing BCL-2 and MCL-1 expression, suggesting that small molecules such as metabolites and lipids are not responsible. Importantly, Proteinase K treatment of supernatant from BH3-mimetic treated cells abolished the ability to upregulate MCL-1 and BCL-2, consistent with the factor(s) being proteinaceous (Fig. 2c). Finally, we investigated whether BH3-only proteins can also have a similar non-cell autonomous effect. HeLa or 293T cells were transfected with tBID and the supernatant was transferred onto recipient cells. Consistent with earlier results, supernatant transferred from tBID transfected cells also caused an up-regulation of BCL-2 and MCL-1 expression (Fig. 2d, Supplementary Fig. 2f). Together, these data demonstrate that BH3-mimetics and BH3-only proteins can upregulate BCL-2 and MCL-1 expression in a non-cell autonomous manner.

**Non-cell autonomous upregulation of anti-apoptotic BCL-2 proteins requires MEK-ERK signalling.** We sought to define the non-cell autonomous mechanism causing anti-apoptotic BCL-2 protein upregulation. For this purpose, HeLa tBID2A cells were treated with venetoclax together with inhibitors targeting pathways previously implicated in anti-apoptotic BCL-2 regulation[14–17]. After co-treatment for 48 h, cell lysates were probed for BCL-2 and MCL-1 expression by western blot. Of all the tested inhibitors, only trametinib (a MEK kinase inhibitor[18]) potently blocked venetoclax induced BCL-2 and MCL-1 expression (Fig 3a, b). The decrease in phosphorylation of ERK1/2, a direct target of MEK[19], validated trametinib activity (Fig. 3b). Upregulation of BCL-2 and MCL-1 was transcriptional, because venetoclax treatment increased RNA

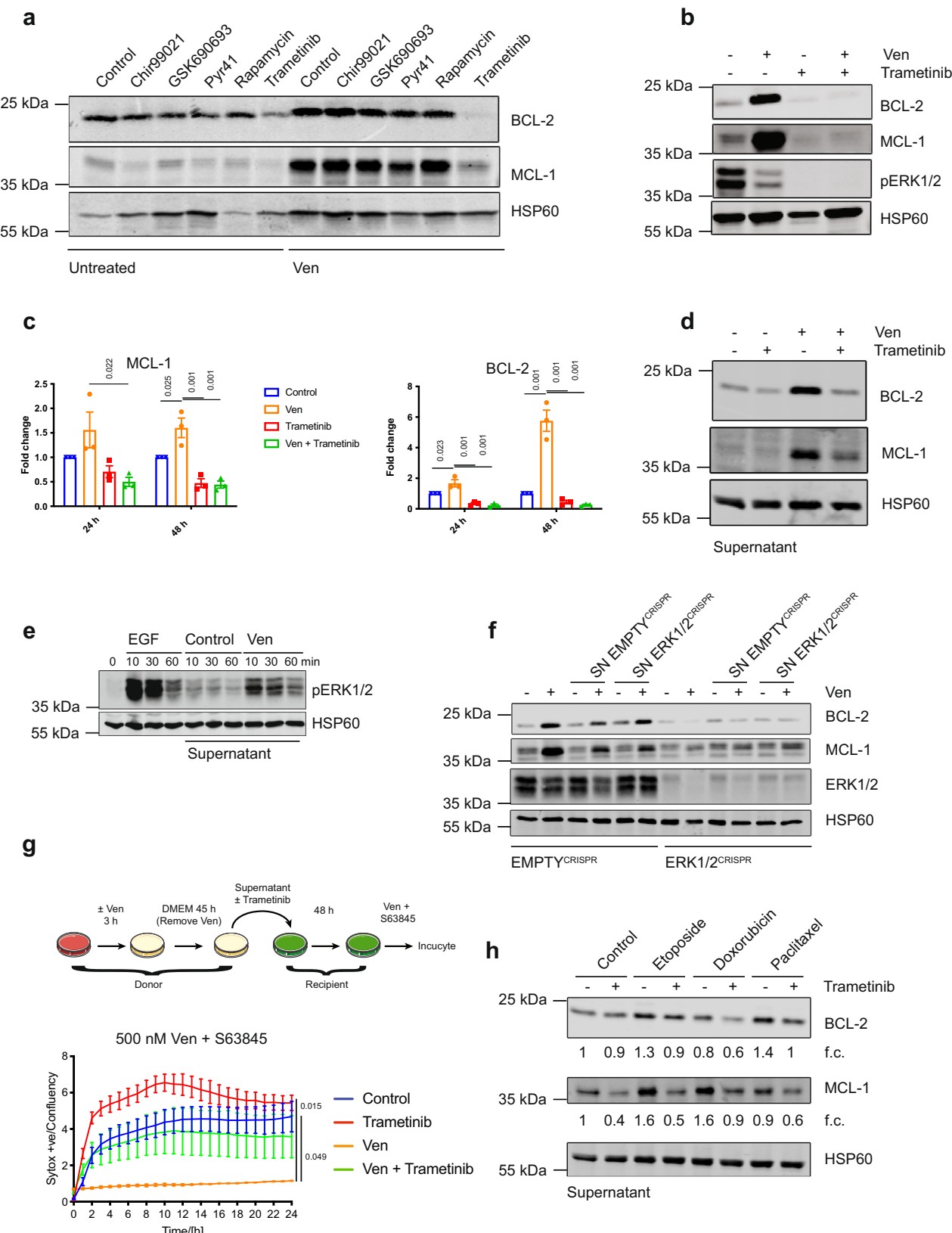

levels, which could be inhibited by trametinib co-treatment (Fig. 3c). The upregulation of MCL-1 by BH3-mimetics was not limited to HeLa cells, since it was also observed in IMR90 lung fibroblasts and CWR-R1 prostate cancer cells (Supplementary Fig. 3a, b). We next investigated whether non-cell autonomous

upregulation of BCL-2 and MCL-1 required MEK-ERK signalling. Trametinib was added to supernatant from venetoclax treated HeLa tBID2A or CWR-R1 cells before the supernatant was added to recipient cells. BCL-2 and MCL-1 upregulation was effectively blocked by trametinib (Fig. 3d, Supplementary Fig. 3c). Again this

**Fig. 3 Non-cell autonomous upregulation of anti-apoptotic BCL-2 proteins requires MEK-ERK signalling. a** HeLa tBID2A cells were untreated or treated with 500 nM venetoclax in combination with the indicated inhibitors for 48 h, harvested and protein expression was analysed by western blot (representative blot of two independent repeats). **b** HeLa tBID2A cells were untreated or treated with 500 nM venetoclax in combination with 500 nM trametinib for 48 h, harvested and protein expression was analysed by western blot (representative blot of three independent repeats). **c** HeLa tBID2A cells were untreated or treated with 500 nM venetoclax in combination with 500 nM trametinib, harvested and RNA expression was analysed by RT-qPCR. $n = 3$ independent experiments; mean values ± s.e.m.; Tukey corrected one-way ANOVA. **d** Supernatant from untreated or venetoclax treated HeLa tBID2A cells was supplemented with 500 nM trametinib before addition onto recipient cells. After 48 h of incubation, recipient cells were harvested and protein expression analysed by western blot (representative blot of three independent repeats). **e** Supernatant from untreated or venetoclax treated HeLa tBID2A cells was added onto recipient cells. After the indicated times, recipient cells were harvested and protein expression analysed by western blot. Treatment with EGF served as a positive control (representative blot of two independent repeats). **f** Control or ERK1/2[CRISPR] HeLa tBID2A cells were treated directly with venetoclax or with supernatant from control or ERK1/2[CRISPR] HeLa tBID2A cells as indicated before harvesting and western blot analysis for protein expression (representative blot of three independent repeats). **g** Supernatant from control or venetoclax treated HeLa tBID2A cells was supplemented with 500 nM trametinib as indicated before addition onto recipient cells. After 48 h of incubation, the recipient cells were treated with 500 nM venetoclax and 500 nM S63845 and survival was monitored by Sytox green exclusion and livecell imaging. $n = 3$ independent experiments; mean values ± s.e.m.; Tukey corrected one-way ANOVA. **h** HeLa cells were treated with the indicated drugs (Etoposide 50 µM, Doxorubicin 2 µM, Paclitaxel 1 µM) for 3 h, followed by 45 h incubation in regular medium. Then the supernatant was harvested, filtered and supplemented or not with trametinib (500 nM) before addition onto recipient cells. After 48 h, cells were harvested and protein expression analysed by western blot. Fold change normalised to loading control is stated below (representative blot of three independent repeats).

effect was transcriptional, because upregulation of MCL-1 RNA in recipient cells was inhibited by trametinib addition to the supernatant (Supplementary Fig. 3d). Furthermore, supernatant of venetoclax treated cells could directly stimulate MEK activity in recipient cells as determined by increased pERK1/2 levels (Fig. 3e). To further investigate these findings, we generated ERK1/2 deficient HeLa tBID2A cells by CRISPR-Cas9 genome editing, hereafter called ERK1/2[CRISPR] cells (Fig. 3f). Media from control or ERK1/2[CRISPR] deleted cells following venetoclax treatment was transferred onto control or ERK1/2[CRISPR] cells, and after 48 h BCL-2 and MCL-1 expression was determined by western blot. BCL-2 and MCL-1 expression increased following incubation with media from venetoclax treated cells or after direct treatment with venetoclax in control cells but was severely attenuated in ERK1/2[CRISPR] cells (Fig. 3f). Finally, we investigated whether MEK signalling, by enabling BCL-2 and MCL-1 upregulation, contributed to venetoclax resistance. HeLa tBID2A cells were incubated with venetoclax ± trametinib for 48 h, after which they were treated with venetoclax and S63845 and cell viability was determined by Sytox Green exclusion and Incucyte live-cell imaging (Supplementary Fig. 3e). Whereas venetoclax pre-treated cells were resistant, trametinib co-treatment completely abolished this resistance, supporting a functional role for MEK signalling in mediating apoptotic resistance via BCL-2 and MCL-1 upregulation. Similarly, transfer of venetoclax treated supernatant conferred resistance to the recipient cells. The resistance was dependent on BCL-2 and MCL-1 upregulation because supplementing the venetoclax treated supernatant with trametinib before addition to recipient cells re-sensitised those cells to the cytotoxic treatment (Fig. 3g). We next investigated if non-cell autonomous upregulation of BCL-2 and MCL-1 was specific to BH3-mimetics or could also be triggered by conventional chemotherapies. Similar to previous experiments, we treated HeLa cells with different chemotherapeutic drugs (etoposide, doxorubicin and paclitaxel) before transferring the supernatant to recipient cells. All three drugs promoted BCL-2 and MCL-1 upregulation, suggesting that apoptosis-inducing stresses can generally induce this effect (Fig. 3h). Supplementing the supernatant with trametinib before addition to recipient cells prevented upregulation, demonstrating that MEK-ERK signalling is essential for the upregulation, consistent with our earlier data. Collectively, these data show that apoptosis-inducing stresses activate MEK-ERK signalling, causing upregulation of BCL-2 and MCL1 and apoptotic resistance in a non-cell autonomous manner.

**FGF signalling mediates non-cell autonomous upregulation of BCL-2 proteins.** Various ligands can bind to receptors that signal

through MEK-ERK, with receptor tyrosine kinases (RTK) being prominent activators of this pathway. Therefore, to determine the paracrine mediator(s) causing BCL-2 and MCL-1 upregulation, we initially focussed on RTK pathways. To identify potential ligands present after BH3-mimetic treatment, supernatant from control or venetoclax treated HeLa tBID2A cells was incubated with a human growth factor antibody array enabling detection of 41 different growth factors (Fig. 4a). Of the growth factors present on the array, FGF2 was increased following venetoclax treatment (Fig. 4a). Upregulation of FGF2 in the supernatant was confirmed by subsequent ELISA analysis (Fig. 4b). Addition of recombinant FGF2 to cells at a concentration similar to what we measured in venetoclax treated supernatant was sufficient to upregulate BCL-2 and MCL-1 expression (Fig. 4c). To directly test the importance of FGF2 in mediating paracrine upregulation of BCL-2 proteins, we generated FGF2 deficient cell lines by CRISPR-Cas9 (Supplementary Fig. 4a). Loss of FGF2 completely suppressed the ability of venetoclax to upregulate BCL-2 and MCL-1 in a paracrine manner, supporting a key role for FGF2 (Fig. 4d). Consistent with activation of FGF-signalling, known target genes of FGF receptors (CDX2[20], DUSP6[21] and SPRY4[22]) were also upregulated in response to supernatant from venetoclax treated cells (Fig. 4e). This upregulation could be blocked by adding trametinib to the supernatant before addition to recipient cells. To investigate the requirement of FGF receptors for upregulation of BCL-2 and MCL-1, we used two different FGFR inhibitors (AZD4547 and PRN1371[23,24]). Co-treatment of HeLa tBID2A cells with either inhibitor and venetoclax prevented BCL-2 and MCL-1 upregulation (Supplementary Fig. 4b). Similarly, supernatant from venetoclax treated HeLa tBID2A cells supplemented with FGFR inhibitors prevented upregulation of BCL-2 and MCL-1 on recipient cells (Fig. 4f). Reduced upregulation of MCL-1 was also observed upon co-treatment of MRC-5 lung fibroblast cells with venetoclax and PRN1371 (Supplementary Fig. 4c). The FGF receptor family is composed of four different receptors[25], of which RNAseq analysis showed that FGFR2 was barely expressed in the HeLa cells used here (Supplementary Fig. 4d). To determine which receptors were responsible to signal the upregulation of BCL-2 and MCL-1 by FGF2, we used RNAi to knock down their expression individually or in combination. Knocking down FGFR1 and FGFR3, either individually or in combination, prevented upregulation of BCL-2 and MCL-1 either after direct treatment (Fig. 4g, Supplementary Fig. 4e) or with venetoclax treated supernatant (Fig. 4h). In contrast, knockdown of FGFR4 failed to affect BCL-2 and MCL-1 expression (Supplementary Fig. 4f–h). We next aimed to understand how FGF2 is regulated

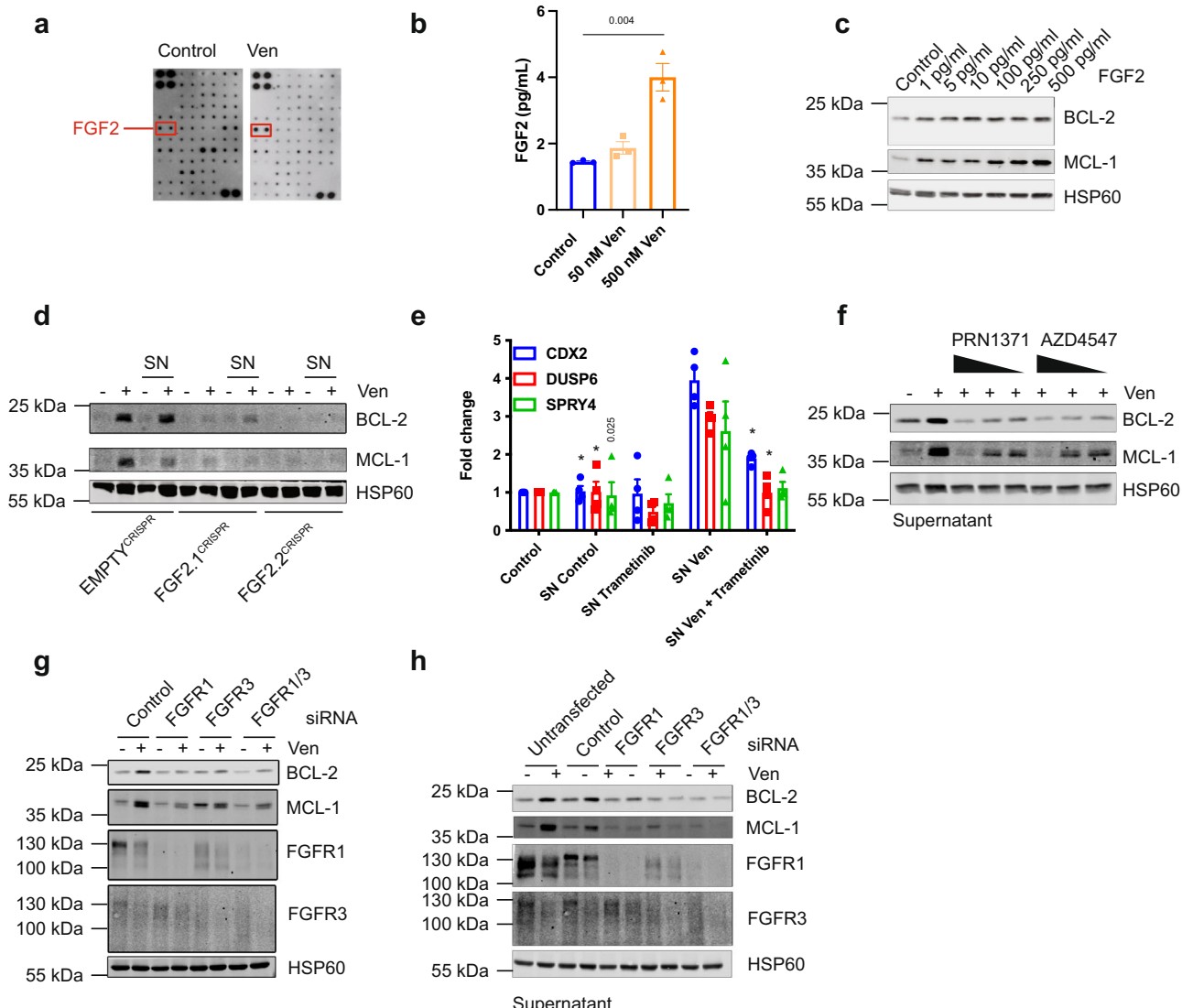

**Fig. 4 FGF signalling mediates non-cell autonomous upregulation of BCL-2 proteins. a** A growth factor membrane ligand array was probed with supernatant from control or venetoclax treated HeLa tBID2A cells. Spots for FGF2 are indicated. **b** Levels of FGF2 were determined in supernatant from control or venetoclax treated HeLa tBID2A cells by ELISA. $n = 3$ independent experiments; mean values ± s.e.m.; unpaired, two-sided $t$-test. **c** HeLa tBID2A cells were treated with recombinant FGF2, harvested after 6 h and protein expression analysed by western blot (representative blot of three independent repeats). **d** Control or FGF2CRISPR HeLa tBID2A cells were directly treated with venetoclax or the respective supernatant for 48 h as indicated before harvesting and analysis of protein expression by western blot (representative blot of three independent repeats). **e** Supernatant from untreated or venetoclax treated HeLa tBID2A cells was supplemented with 500 nM trametinib before addition onto recipient cells. After 3 h, recipient cells were harvested and RNA expression of FGF target genes analysed by RT-qPCR. $n = 4$ independent experiments; mean values ± s.e.m.; *$p < 0.0001$ compared to SN Ven; Dunnetts corrected one-way ANOVA. **f** Supernatant from control or venetoclax treated HeLa tBID2A cells was supplemented with decreasing doses of FGFR inhibitors as indicated (AZD: 5 μM, 2.5 μM, 1.25 μM; PRN1371: 10 μM, 5 μM, 2.5 μM) before addition onto recipient cells for 48 h and analysis of protein expression by western blot (representative blot of three independent repeats). **g** HeLa tBID2A cells were transfected with siRNA targeting FGFR1 and FGFR3 alone or in combination for 24 h before addition of 500 nM venetoclax, harvesting after 48 h and analysis of protein expression by western blot (representative blot of three independent repeats). **h** HeLa tBID2A cells were transfected with siRNA targeting FGFR1 and FGFR3 alone or in combination for 24 h before addition of control or venetoclax treated supernatant from control cells, harvesting after 48 h and analysis of protein expression by western blot (representative blot of three independent repeats).

in response to BH3-mimetic treatment. A minor, yet significant increase in FGF2 mRNA was detected in both wild type and BAX BAK deficient HeLa tBID2A cells following venetoclax treatment (Fig. 5a), however there was a decrease in FGF2 protein levels over time (Fig. 5b). Given the lack of increase in FGF2 protein level, we investigated whether inhibiting transcription in donor cells affected the upregulation of BCL-2 and MCL-1 in recipient cells. Supernatant from cells co-treated with venetoclax and actinomycin D still led to upregulation of MCL-1 and BCL-2 after

transfer (Fig. 5c), suggesting that transcriptional regulation is not necessary. In contrast, actinomycin D prevented upregulation of MCL1 in supernatant-treated recipient cells (Fig. 5d), corroborating transcriptional upregulation of MCL-1 in recipient cells. These experiments demonstrate that the activation of FGF2 in response to apoptotic stress is independent of induced expression, but instead may be due to increased release from the cell. Collectively, these data demonstrate that FGF-signalling, triggered by release of FGF2 from BH3-mimetic treated cells, is required and

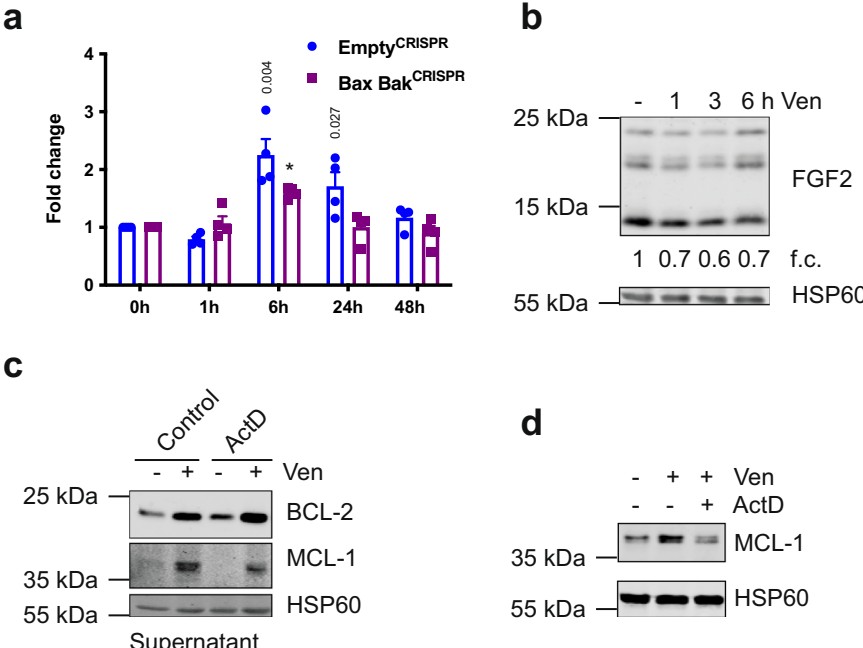

**Fig. 5 Non-cell autonomous upregulation of BCL-2 proteins is independent of transcription. a** HeLa tBID2A cells were treated with 500 nM venetoclax for the indicated time, RNA was harvested and expression of FGF2 mRNA analysed by qPCR. $n = 4$ independent experiments; mean values ± s.e.m.; *$p < 0.0001$; unpaired, two-sided $t$-test. **b** HeLa tBID2A cells were treated with 500 nM venetoclax for the indicated time, harvested and protein expression analysed by western blot. Fold change normalised to loading control is stated below (representative blot of three independent repeats). **c** HeLa tBID2A cells were treated with venetoclax ± actinomycin D for 3 h, followed by 45 h incubation in regular medium. Then the supernatant was harvested, filtered and added onto recipient cells. After 48 h, recipient cells were harvested and protein expression analysed by western blot (representative blot of three independent repeats). **d** HeLa tBID2A cells were treated with venetoclax for 3 h, followed by 45 h incubation in regular medium. Then the supernatant was harvested, filtered, supplemented with actinomycin D as indicated and added onto recipient cells. After 3 h, recipient cells were harvested and protein expression analysed by western blot (representative blot of three independent repeats).

sufficient for non-cell autonomous upregulation of anti-apoptotic BCL-2 proteins.

**FGF signalling is essential for non-cell autonomous apoptotic resistance.** We next investigated the contribution of FGF-mediated BCL-2 and MCL-1 upregulation to apoptosis resistance. First, HeLa tBID2A cells were co-treated with venetoclax and inhibitors of FGF signalling (PRN1371, AZD4547) or MEK (trametinib). Whereas the venetoclax treated cells were resistant to apoptosis induced by re-treatment with venetoclax and S63845, cells co-treated with FGF signalling inhibitors were re-sensitised to apoptosis (Fig. 6a, Supplementary Fig. 5a). As observed previously, increasing the dose of venetoclax and S63845 could restore sensitivity to venetoclax pretreated cells. Next, the potential of FGF signalling to promote non-cell autonomous apoptotic resistance was investigated. Supernatant was harvested from venetoclax treated HeLa tBID2A cells and supplemented with inhibitors of the FGF signalling pathway before addition onto recipient cells. Consistent with our earlier data, supernatant from venetoclax treated cells conferred apoptotic resistance to recipient cells (Fig. 6b, Supplementary Fig. 5b). Crucially, apoptotic sensitivity was restored upon inhibition of either FGF or MEK signalling. Again, increasing the dose of venetoclax and S63845 could kill the venetoclax pre-treated resistant cells. To investigate this further, we assessed effects on long-term clonogenic survival. Supporting earlier data, supernatant from venetoclax treated cells could confer long-term protection from apoptosis, which was restored by supplementation of FGF signalling inhibitors (Fig. 6c). To further investigate these findings, we transferred supernatant from venetoclax treated control or FGF2 deficient cells

(Supplementary Fig. 4a) and tested survival in response to venetoclax and S63845. While the supernatant from the wild-type cells protected from apoptosis, this protection was attenuated in cells treated with supernatant from FGF2 deficient cells (Fig. 6d). Increasing the concentration of venetoclax and S63845 could overcome the protective effect of supernatant from venetoclax treated cells. These data support a model whereby following apoptotic stress, cells can signal non-cell autonomous apoptotic resistance by FGF-signalling dependent on the upregulation of BCL-2 and MCL-1 (Fig. 6e). Finally, we investigated whether a similar mechanism may be evident in cancer. Interrogating a total of 25 different TCGA cancer datasets, we first removed patients with mutations in components of the FGFR signalling pathway. These patients might show an altered activation of the FGF signalling pathway independent of the non-cell autonomous pathway described in this work. We next calculated an FGF pathway activation score by determining the mean expression of ten FGF target genes as a proxy for activation of FGF signalling. The FGF score was then correlated with MCL-1 and BCL-2 expression to determine if increased activation of FGF signalling correlates with increased levels of prosurvival BCL-2 family gene expression. Using this approach, we identified several cancer types that displayed a correlation between FGF activation and BCL-2 and/or MCL-1 expression (Fig. 6f, Supplementary Fig. 6a, b). Next, we determined whether this correlation had an influence on disease progression by stratifying patients into groups based on FGF score and BCL-2 or MCL-1 expression. Strikingly, in three out of 25 tested cancer types, survival of the high scoring group (high activation of FGF signalling and high expression of BCL-2 or MCL-1) was significantly decreased when compared to the

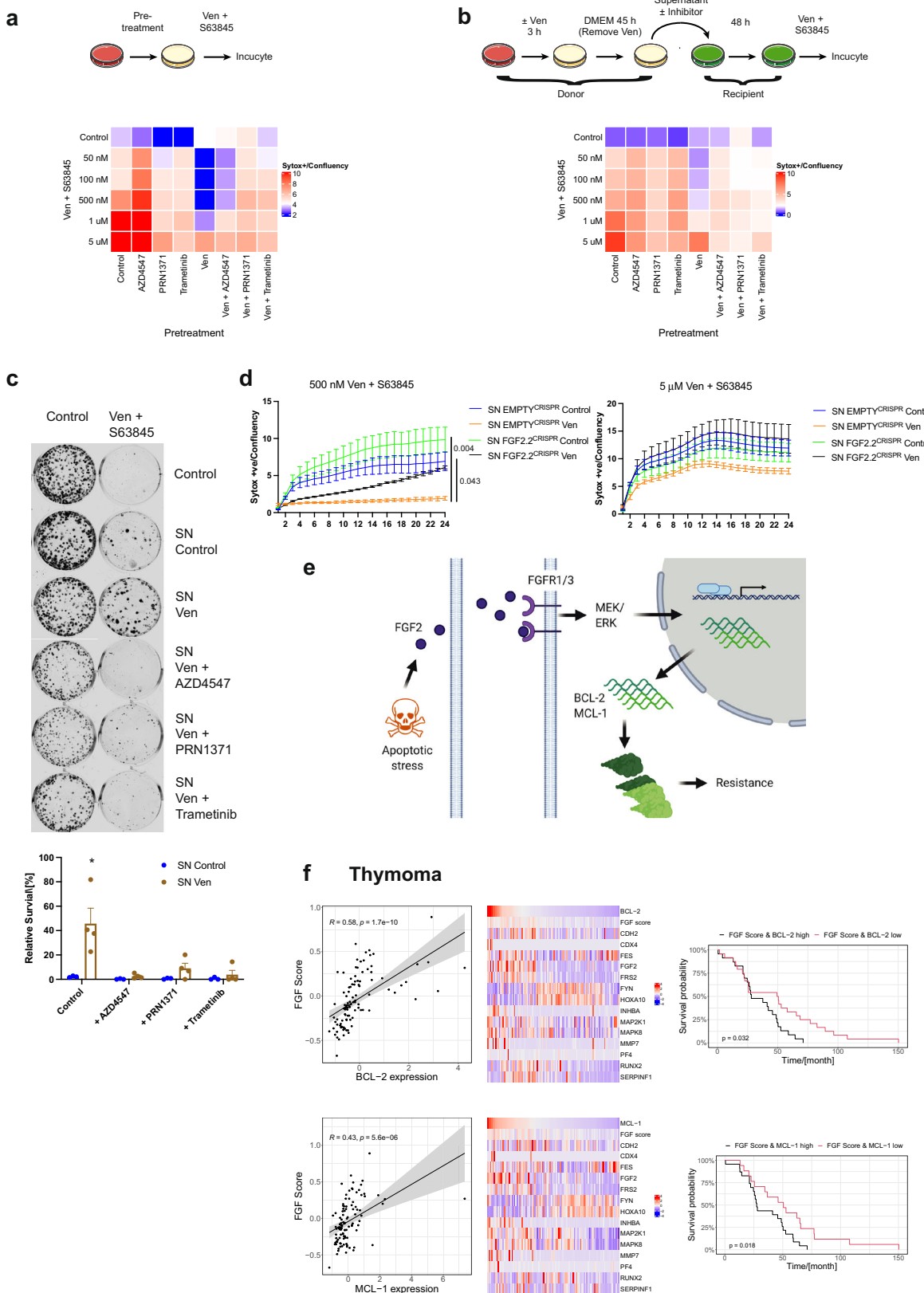

low scoring group (Fig. 6f, Supplementary Fig. 6a, b). These data suggest that FGF-mediated resistance might have a protective effect on cancer cell survival and therefore worsen prognosis.

**FGF signalling induces MCL-1 in the skin and regulates wound healing dynamics.** Oncogenic processes are often subverted homoeostatic mechanisms. We therefore sought to define physiological roles for FGF-signalling induced anti-apoptotic BCL-2

**Fig. 6 FGF signalling is essential for non-cell autonomous apoptotic resistance. a** HeLa tBID2A cells were treated with 500 nM venetoclax in combination with RTK pathway inhibitors (AZD4547 (5 μM), PRN1371 (10 μM) or trametinib (500 nM)) as indicated for 48 h. Then the cells were treated with the indicated concentrations venetoclax + S63845 and cell survival was monitored by Incucyte. Heatmap colours show cell death at 24 h (mean from $n = 3$ independent experiments), corresponding survival curves are shown in Supplementary Fig. 5a. **b** HeLa tBID2A cells were incubated with 500 nM venetoclax treated supernatant supplemented with RTK pathway inhibitors (AZD4547 (5 μM), PRN1371 (10 μM) or trametinib (500 nM)) as indicated before addition onto recipient cells for 48 h. Then the cells were treated with the indicated concentrations venetoclax + S63845 and cell survival was monitored by Incucyte. Heatmap colours show cell death at 24 h (mean from $n = 3$ independent experiments), corresponding survival curves are shown in Supplementary Fig. 5b. **c** HeLa tBID2A cells were plated at low density (1000 cells per 6 well) and incubated with 500 nM venetoclax-treated supernatant supplemented with RTK pathway inhibitors (AZD4547 (5 μM), PRN1371 (10 μM) or trametinib (500 nM)) as indicated for 48 h. Cells were then treated with 2.5 μM venetoclax + S63845 for another 48 h before replacement with regular growth medium. After an additional 5 days colonies were visualised by crystal violet staining and quantified. $n \geq 3$ independent experiments; mean values ± s.e.m.; *$p < 0.05$ compared to all other treatments; Tukey corrected one-way ANOVA. **d** Control or FGF2 deficient cells were treated for 3 h with 500 nM venetoclax followed by 45 h regular medium. Then the supernatant was harvested and added to control cells for 48 h. After that, the cells were treated with venetoclax + S63845 with the indicated doses and survival monitored by Incucyte. $n = 3$ independent experiments; mean values ± s.e.m.; *$p < 0.05$ compared to venetoclax treatment at 24 h; Tukey corrected one-way ANOVA. **e** Working model. **f** Pearson correlation between FGF Score and BCL-2 (upper left) or MCL-1 (lower left) in the TCGA Thymoma dataset (± s.e.m, 105 patients). FGF score, FGF receptor target gene and BCL-2 (upper middle) or MCL-1 (lower middle) expression in the TCGA Thymoma dataset. Survival of TCGA Thymoma patients stratified by FGF score and BCL-2 (upper right) or MCL-1 (lower right) expression ($p$ value was calculated with a log-rank test).

expression. Because our data are consistent with a model in which stressed cells can alert the microenvironment to increase their threshold against apoptosis, we decided to test this hypothesis in a physiologically relevant and experimentally tractable model. In the epidermis, apoptosis has been found to play an important role in skin repair[26,27]. Notably, re-epithelization of the skin relies heavily on FGF signalling[28]. We therefore set out to examine whether FGF signal transduction might also modulate MCL-1 expression in the skin upon wound infliction. We hypothesised that an injured tissue might increase its apoptotic threshold to limit damage due to cell death in areas that require regeneration. This increased resistance to apoptosis could potentially protect against excessive cell death which could hinder tissue regeneration. Indeed, subjecting the dorsal skin of mice to 1 cm$^2$ full-thickness excisional wounds promoted upregulation of MCL-1 in keratinocytes in the vicinity of the wound (Fig. 7a). We next examined whether inhibition of FGF signalling affected MCL-1 expression by utilising the FGFR inhibitor AZD4547 and the MEK inhibitor trametinib. We administered the inhibitors for four consecutive days by subcutaneous injection prior to infliction of the wound and harvested the skin three days post wound infliction (PWI). Our results indicate that administration of either inhibitor dramatically decreased the number of MCL-1 + cells following injury (Fig. 7b, c), suggesting that upregulation of MCL-1 in response to skin injury is dependent on FGF signalling.

To investigate the physiological role of FGF mediated MCL-1 upregulation, we monitored wound closure dynamics. For this aim, we inflicted full-excisional wounds on dorsal skins of control or inhibitor treated mice and evaluated wound coverage at specific time points PWI (Supplementary Fig. 7a). In control mice, the wound size was reduced by ~40% after just one day, whereas in inhibitor treated mice no coverage was seen at this time point (Fig. 7d). This delay was accompanied by a decreased area of re-epithelization (Fig. 7d, Supplementary Fig. 7b, c). To examine the contribution of epidermal keratinocytes to the observed phenotypes, we next evaluated proliferation in the wound bed. We harvested wounds at three and seven days PWI. FGFR and MEK-inhibition resulted in decreased proliferation evident by Mcm2 and Ki67 immunostaining (Supplementary Fig. 7b, d, e). The attenuation in wound closure phenotypes seen upon FGFR-inhibition could potentially be facilitated via decreased basal cell expansion and suprabasal migration capacity. Elegant studies have shown that in early stages of wounding healing the leading edge of the wound is mostly composed of non-proliferative migratory cells that can drive rapid re-

epithelialization[29,30]. Our analysis revealed that FGFR-inhibited wounds displayed a less pronounced leading edge by seven days PWI, when compared to control (Supplementary Fig. 7b, c). These findings suggest that suprabasal keratinocytes may also contribute to the attenuated wound healing phenotypes seen upon inhibition of FGF receptors and MEK. Our data indicate that FGF signalling induces the expression of MCL-1 in the skin and affects the contribution of both basal and suprabasal keratinocytes to the wound repair process. Overall, our results suggest a mechanism in which cells protect their tissue integrity by increasing the apoptotic threshold in response to stress by FGF2 mediated upregulation of pro-survival proteins.

## Discussion

Innate or acquired resistance to cell death is of fundamental importance in health and disease. For instance, evasion of apoptosis can both promote cancer and inhibit treatment response, leading to tumour relapse[31]. To understand how cells can resist cell death, we used BCL-2 targeting BH3-mimetics as tool compounds. Unexpectedly, we uncovered a non-cell autonomous mechanism that enables apoptotic resistance. We found that upon apoptotic stress, cells can release the growth factor FGF2. By activating MEK-ERK signalling, FGF2 upregulates anti-apoptotic BCL-2 protein expression[32,33] in neighbouring cells, protecting against apoptosis. Importantly, resistance can be overcome by co-treatment of BH3-mimetics with FGFR-inhibitors, demonstrating the functional significance of the pathway. Accordingly, in an in vivo injury model inhibition of FGFR signalling prevents MCL-1 upregulation, apoptosis and successively wound healing. Finally, we describe a correlation between increased FGF signalling, anti-apoptotic BCL-2 protein expression and poor patient prognosis, suggesting its relevance in vivo. As we discuss further, the process we describe here, where cell death promotes life, may have various pathophysiological functions.

Most cancer therapies work by killing tumour cells, consequently resistance to cell death profoundly impacts on therapeutic efficacy[31]. Typically, cancer cells can evade apoptosis through inactivating mutations in pathways that sense damage or activate cell death[31]. For instance, in chronic lymphocytic leukaemia (CLL), BCL-2 mutations have recently been reported that render BCL-2 unable to effectively bind the BH3-mimetic venetoclax, causing apoptotic resistance[34–37]. The resistance mechanism we report here is not mutation based, but instead is due to dying cells releasing FGF2 that causes transient apoptotic resistance in

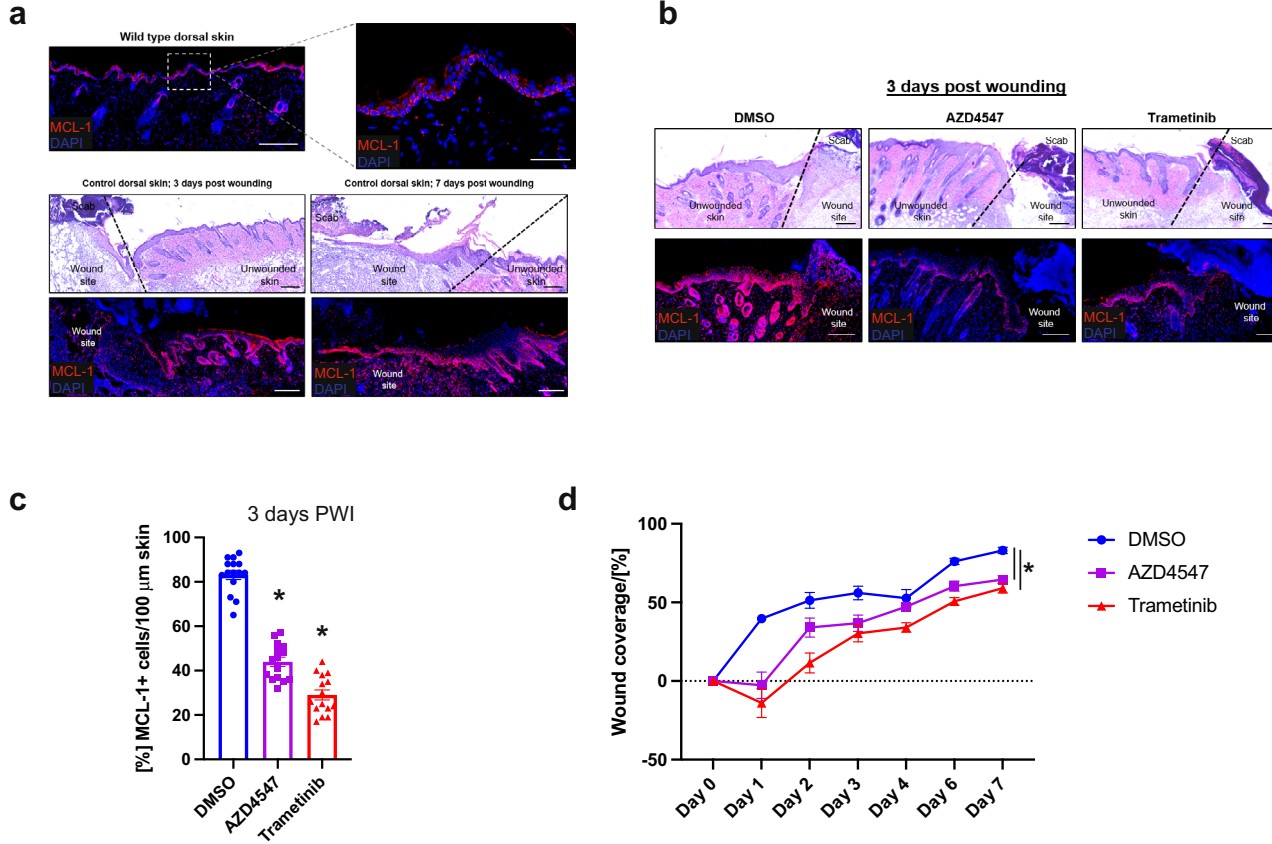

**Fig. 7 FGFR and MEK inhibition decreases MCL-1 expression and hinders wound repair. a** H & E and immunostaining of MCL-1 in unwounded skins and 3 or 7 days post wounding (PWI). Representative image of $n = 15$ mice, Scale bar: 200 μm, Inset 50 μm. The wound edge is indicated with a black broken line. **b** H & E and immunostaining of MCL-1 in skins 3 days post wounding in DMSO or inhibitor treated mice as indicated. Representative image of $n = 15$ mice per group, Scale bar: 200 μm. The wound edge is indicated with a black broken line. **c** Quantification of MCL-1 positive cells in the area surrounding the wound site. $n = 15$ mice per group, each dot represents an individual mouse, mean values ± s.e.m.; *$p < 0.0001$ compared to DMSO; Tukey corrected one-way ANOVA. **d** Quantification of dorsal wound coverage in DMSO or inhibitor treated mice over a 7-day period. $N \geq 2$ mice per group, each dot represents an individual mouse, mean values ± s.e.m.; Tukey corrected one-way ANOVA at day 7.

surrounding cells. This mechanism fits the concept of persistence, which has emerged to explain transient, non-heritable resistance of tumour cells to therapy[38]. Cancer cells that transiently evade cell death, called persister cells, blunt the effectiveness of chemotherapy and provide a cell pool from which drug-resistant tumours may arise. The mechanism we describe herein represents a non-cell autonomous way of generating persistence (via FGF2 release) that requires activation of the core apoptotic pathway. Suggestive of its relevance in vivo, we find that in certain cancer types there is a correlation between FGF-signalling, anti-apoptotic BCL-2 protein expression and poor prognosis. Importantly, we find that inhibition of FGF receptor signalling or of downstream MEK-ERK signalling greatly impedes the ability of stressed cells to promote survival in a non-cell autonomous manner. Our data show that apoptotic resistance is transient lasting multiple days, after which cells become sensitised again to BH3-mimetics. Extrapolating these findings to a clinical setting, one possibility may be to combine apoptosis inducing cancer therapy with FGFR inhibitors. Additionally, our data suggest that intermittent dosing of apoptosis inducing therapies, employing so-called drug holidays, may help circumvent apoptotic resistance.

Some molecular and cellular mechanisms are known to be shared by both wounds and cancers, bringing forward the concept of tumours as over-healing[39]. Tissue integrity is essential for the proper functioning of any multicellular organism. To maintain this integrity, breaches have to be refilled, for instance by increased migration and proliferation. Epithelial tissue like skin is often subjected to injury, and efficient repair is necessary to close the wound and to restore proper barrier functions. This wound repair is facilitated by growth factors like FGF2 to promote cell proliferation and angiogenesis[40,41]. Interestingly, secretion of FGF2 was shown to be induced by shear stress[42], and activation of ERK signalling is observed in several instances of tissue regeneration[43,44]. Pro-survival ERK signalling engaged by EGF released by apoptotic cells has recently been found to promote tissue integrity[45,46]. Importantly, while we find that FGF2 can be released from viable cells under conditions of apoptotic stress, EGF release required apoptosis. This highlights that pro-apoptotic signalling can engage multiple pro-survival mechanisms. The role of BCL-2 proteins in skin injury has not been extensively studied, however MCL-1 seems to be involved in keratinocyte survival and differentiation[47]. Our data now provide a potential connection between FGF2 and MCL-1 in wound healing: In response to cell stress due to injury, cells secrete FGF2 to induce pro-survival MCL-1 expression in neighbouring cells and to increase their apoptotic threshold. Increasing resistance to death in this setting might have several reasons, for example to limit the sustained damage by preventing extensive cell death or to promote regeneration of the tissue by protecting heavily proliferating cells from death. Inhibiting FGF signalling during wounding or decreasing the apoptotic threshold with BH3-mimetics therefore delays wound healing. An intriguing avenue

for the future would be to examine how signals released from apoptotic cells can be harnessed to facilitate tissue regeneration.

A remaining question is how apoptotic stress leads to FGF2 release in dying cells. FGF2 is secreted from cells in a non-conventional manner that remains to be fully elucidated[48]. Importantly, while FGF2 release occurs during caspase-dependent apoptosis, it is not dependent on apoptosis - neither inhibition of caspase function nor MOMP prevents FGF2 release. This suggests that neutralisation of anti-apoptotic BCL-2 proteins exerts a non-lethal signalling function, leading to FGF2 release. Indeed, a variety of non-apoptotic functions for BCL-2 have been reported previously, for instance roles in calcium signalling or metabolism[49]. Alternatively, BH3-only proteins may be responsible for this FGF2 release. FGF2 is secreted in a non-canonical manner, which involves its binding to $PI(4,5)P_2$ at the plasma membrane followed by insertion of FGF2 oligomers through the membrane[50]. Although tBID was described to be able to interact with various membranes like artificial liposomes[51], lysosomes[52] or mitochondria[53], it remains to be determined if this function of tBID can be extended to FGF2 secretion.

Our data further emphasise that cell death exerts a plethora of non-cell autonomous effects. These include context dependent pro-proliferative, inflammatory and apoptotic activities[54–56]. The mechanism we describe here represents an FGF-driven pro-survival effect of apoptotic stress. As discussed, this effect may have important implications for dictating therapeutic efficacy of apoptosis-inducing cancer therapy. Beyond this, we show that apoptotic stress induced survival signalling may also have a physiological role linking tissue stress to tissue repair.

## Methods

**Cell lines and chemicals**. HeLa and 293T cells were purchased from ATCC, HeLa tBID2A BCL-2 were previously described[9], MRC5 and IMR90 cells were a gift from Peter Adams, Beatson Institute, and CWR-R1 cells were a gift from Arnaud Blomme, Beatson Institute. Cell lines were not authenticated. Cells were regularly tested negative for mycoplasm. HeLa, HeLa tBID2A BCL-2 cells[9], IMR90, MRC-5 and 293 T cells were cultured in DMEM high-glucose medium supplemented with 10% FCS and 2 mM glutamine. CWR-R1 cells were cultured in RPMI high-glucose medium supplemented with 10% FCS and 2 mM glutamine. MRC5 and IMR90 cells were cultured in 3% oxygen. To select for venetoclax resistant cells, HeLa tBID2A BCL-2 cells were cultured continuously in the indicated dose of venetoclax for 14 days or cultured for 8 h in venetoclax followed by 16 h normal medium daily. The following drugs and chemicals were used: ABT-199/venetoclax (AdooQ BioScience, A12500-50), ABT-263/Navitoclax (ApexBio, A3007), ABT-737 (ApexBio, A8193), Actinomycin D (Calbiochem, 114666), AZD4547 (Selleck, S2801), Chir99021 (GSK3 inhibitor, final concentration 3 μM, Gift from D. Murphy), Cycloheximide (Sigma, 1810), Doxorubiucin (Sigma D1515), EGF (Sigma, E4127), Etoposide (Sigma, E1383), FGF2 (Thermo, PHG0263), GSK690693 (AKT inhibitor, final concentration 1 uM, gift from Daniel Murphy, University of Glasgow), Paclitaxel (Sigma, T7191), PRN1371 (Selleck, S8578), Propidium iodide (Sigma, P4170), Proteinase K (Thermo, 25530049), Pyr41 (E1 Ubiquitin ligase inhibitor, final concentration 50 μM, Sigma, N2915), qVD-OPh (AdooQ BioScience, A14915-25), Rapamycin (mTOR inhibitor, final concentration 1 μM, Santa Cruz, sc-3504), S55746 (ProbeChem, PC-63502), S63845 (Chemgood, C-1370), Sytox Green (Thermo, S7020), Syto 21 (Sigma, S7556) and trametinib (MEK inhibitor, Cambridge Bioscience, HY-10999).

**Lentiviral transduction**. CRISPR-Cas9-based genome editing was performed using LentiCRISPRv2-puro (Addgene #52961) or LentiCRISPRv2-blasti[9] using the following guide sequences: hBAX, 5′-AGTAGAAAAGGGCGACAACC-3′; hBAK, 5′-GCCATGCTGGTAGACGTGTA-3′; hERK1, 5′-CAGAATATGTGGCCACA CGT-3′; hERK2, 5′-AGTAGGTCTGATGTTCGAAG-3′; hFGF2.1, 5′-TATGCA AGTCCAACGCACTG -3′ and hFGF2.2, 5′-CGAGCTACTTAGACGCACCC-3′.

For stable cell line generation, 5*10^6 293FT cells were transfected in 10 cm dishes using 4 μg polyethylenimine (PEI, Polysciences) per μg plasmid DNA with lentiCRISPR_V2 (Addgene 52961): gag/pol (Addgene 14887): pUVSVG (Addgene 8454) at a 4:2:1 ratio. After 48 and 72 h of transfection, virus containing supernatant was filtered, supplemented with 1 μg/ml polybrene (Sigma) and added to 50.000 recipient cells in a 6 well plate. Selection with appropriate antibiotics (1 μg/ml puromycin (Sigma) or 10 μg/ml blasticidin (InvivoGen)) was started 24 h after the last infection and continued for one week[57].

**Supernatant assays**. Generally, cells were treated for 3 h with 500 nM venetoclax, then the medium was replaced with regular growth medium. After an additional 45 h the supernatant was harvested, filtered and added onto recipient cells. For Proteinase K treatment, supernatant was treated with 200 μg/ml Proteinase K for 60 min at 50 °C, followed by 5 min at 95 °C. After cooling down, the treated supernatant was added to recipient cells. For centrifugal filtration, supernatant was filtered and added into an Amicon Ultra 15 ml tube (Merck) with a 3 kDa cut-off. After spinning at 4000 g for 60 min, the concentrate was diluted with regular growth medium to its original volume and the concentrate or the flowthrough was added onto recipient cells. The FGF2 ELISA was performed using the Human FGF-basic ELISA MAX Deluxe (Biolegend) according to the manufacturer's instructions after 50x concentration of the supernatant by centrifugal filtration (see above). The final FGF2 concentration was determined using a standard of recombinant FGF2, taking into account the concentration step.

**Plasmid and siRNA transfection**. For plasmid transfection, Lipofectamine 2000 was used according to the manufacturer's instructions. Transfection of siRNA was performed using Oligofectamine according to the manufacturer's instructions. The following siGENOME SMARTpool siRNAs from Dharmacon were used: Non-targeting, D0012061305; hFGFR1, M-003131-03-0005; hFGFR3, M-003133-01-0005 and hFGFR4, M-003134-02-0005.

**Western blotting**. Cell lysates were prepared using lysis buffer (1% NP-40, 0.1% SDS, 1 mM EDTA, 150 mM NaCl, 50 mM Tris pH7.5, supplemented with Complete Protease Inhibitors (Roche) and PhosSTOP (Roche)). Protein concentration was determined using Bio-Rad Protein Assay Dye (5000006), and lysates were separated by SDS-PAGE followed by blotting onto nitrocellulose membranes and incubation with primary antibody (1:1000 in 5% milk) overnight. After washing in TBS/T and incubation with secondary antibody (Li-Cor IRDye 800CW donkey anti-rabbit, 926-32213, dilution 1:20000), blots were developed on a Li-Cor Odyssey CLx system and acquired using Imagestudio (Li-Cor). The following primary antibodies were used: Actin (A4700, Sigma), BAK (12105, Cell Signaling), BAX (2772, Cell Signaling), BCL-2 (4223, Cell Signaling), ERK1/2 (4695, Cell Signaling), basic FGF (20102, Cell Signaling), FGFR1 (9740, Cell Signaling), FGFR3 (4574, Cell Signaling), FGFR4 (8562, Cell Signaling), GFP (In house), HSP60 (4870, Cell Signaling), MCL-1 (5453, Cell Signaling), pERK1/2 (4370, Cell Signaling), Caspase 3 (9662, Cell Signaling), Caspase 9 (9502, Cell Signaling), PARP1 (9532, Cell Signaling) alpha-Tubulin (T5168, Sigma) and cleaved Caspase 3 (9664, Cell Signaling).

**Quantitative RT-PCR**. RNA from cultured cells was isolated with the GeneJET RNA purification kit according to the manufacturer's instructions. cDNA synthesis was performed according to the manufacturer's instructions using the High Capacity cDNA Reverse Transcription Kit (Thermo), and qPCR was performed with the Brilliant III Ultra-Fast SYBR Green qPCR Master Mix (Agilent Technologies) on a QuantStudio 3 machine (Applied Biosystems) with the following programme: 3 min at 95 °C, 40 cycles of 20 s at 95 °C, 30 s at 57 °C, 30 s at 72 °C and a final 5 min at 72 °C. Results were analysed using the $2^{-\Delta\Delta Ct}$ method. Primer sequences can be found in Supplementary Table 1.

**Cell death assays**. Short-term cell death was determined with an Incucyte FLR or Zoom imaging system (Essen Bioscience)[58]. Cells were treated as indicated in the Figure legend together with 30 nM Sytox green and imaged every 1 or 2 h. Analysis was performed using the Incucyte software and the number of dead (Sytox green positive) cells was normalised to the confluency at $t = 0$. Alternatively, cells were pre-treated for 1 h with 50 nM Syto 21, followed by cytotoxic treatments as indicated together with 5 μg/ml propidium iodide. Long-term colony formation assay was performed by plating 1000 cells per 6 well and treatment as described in the Figure legend. After 48 h of treatment with supernatant, the medium was changed to 2.5 μM venetoclax/S63845. Medium was changed to regular growth medium after an additional 48 h, and resulting colonies were stained with crystal violet after an additional 5 days. Cell death analysis via FACS was performed using Annexin V - propidium iodide staining[59]. In short, treated cells were harvested and stained with 5 μg/ml propidium iodide and Annexin V (Biolegend) in Annexin V-binding buffer (10 mM Hepes pH 7.4, 140 mM NaCl, 2.5 mM $CaCl_2$) for 15 min. Flow cytometry was conducted on a BD FACSCalibur machine with BD CellQuest software and analysed using Flowing software; cells negative for propidium iodide and Annexin V were considered alive.

**Membrane ligand array**. Supernatant was harvested from HeLa tBID2A cells as described above and used to probe a Human Growth Factor Antibody Array (Abcam, ab134002) according to the manufacturer's instructions.

**Bioinformatic analysis**. Mutation, survival and gene expression data (Thymoma TCGA Firehose Legacy) was downloaded from cBioPortal (www.cbioportal.org). Samples with alterations (Deletions, mutations or amplifications) in components of the FGFR signalling pathway (FGFR1, FGFR3, GRB2, FRS2, SOS1, HRAS, RAF1, MAPK1, MAPK3) were removed. The FGF score was determined by averaging the

expression of the *FGF2* induced gene set (CDH2, CDX4, FES, FGF2, FRS2, FYN, HOXA10, INHBA, MAP2K1, MAPK8, MMP7, PF4, RUNX2, SERPINF1) from the Harmonizome database (https://maayanlab.cloud/Harmonizome/gene_set/fgf2induced/GeneRIF+Biological+Term+Annotations). Pearson correlation was calculated between FGF score and BCL-2 or MCL-1 expression. Samples were stratified into thirds based on FGF score, BCL-2 or MCL-1 expression. Survival was analysed comparing samples with high FGF score and high expression of BCL-2 or MCL-1 with low FGF score and low expression of BCL-2 and MCL-1. The used code is available as a supplementary software file (Supplementary Software 1).

**Wound healing experiments**. For all wound repair experiments, C57BL/6 J mice were sex and age matched (8-weeks-old) and randomly assigned to different treatment groups. Mice were first shaved and full-thickness 1 cm$^2$ excision wounds were performed on the dorsal skin. Following wound infliction, mice were euthanized with $CO_2$ at various time points and wounded skins were embedded in OCT for analysis. Housing, care and wounding experiments were approved by the ethical committee of the Technion - Israel Institute of Technology

**Immunofluorescence and Hematoxylin and Eosin staining**. Skins frozen in OCT were sectioned at 12 µm and fixed in 4% paraformaldehyde for 20 min at room temperature. Samples were then blocked for 2 h, followed by incubation with primary antibodies diluted in blocking buffer overnight at 4 °C. Following washing, samples were incubated with secondary antibodies (Alexa Fluor 488/546/633, Thermo) for 1 h at room temperature. The following primary antibodies were used: MCL-1 (1:800, 5453, Cell Signaling), Ki67 (1:100, 9882, eBioscience), Mcm2 (1:500, 4461, Abcam), active ItgB1 (1:100, BD, 550531) and Alexa Fluor 568 phalloidin. Sample analysis was performed on a Zeiss LSM 880 confocal microscope and analysed using Zen software. Sections were first treated with Hematoxylin followed by $H_2O$, differentiator (0.3% alcoholic HCl), $H_2O$, 95% EtOH, Eosin, 95% EtOH, twice in 100% EtOH and three times in xylene before mounting in xylene based mounting medium.

**Reporting summary**. Further information on research design is available in the Nature Research Reporting Summary linked to this article.

## Data availability
Source data are provided with this paper. The TCGA data used in this study are available in the cBioPortal database under [www.cbioportal.org/study/summary?id=thym_tcga], [https://www.cbioportal.org/study/summary?id=kich_tcga] and [https://www.cbioportal.org/study/summary?id=ucec_tcga]. The FGF score was based on the fgf2induced dataset from the Harmonizome [https://maayanlab.cloud/Harmonizome/gene_set/fgf2induced/GeneRIF+Biological+Term+Annotations]. The remaining data are available within the Article and Supplementary Information. Source data are provided with this paper.

## Code availability
Bioinformatic analysis was conducted in R, the used code is available as supplementary software file.

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

## Acknowledgements

This work was supported by Cancer Research UK grant C40872/A2014 (S.W.G.T), CRUK core funding A29799 to K.B. and a Tenovus small pilot grant (F.J.B). We thank Joel Riley and Catherine Winchester for reviewing the manuscript and all members of the Tait laboratory for helpful suggestions as well as Hasan Uludağ (University of Alberta, Canada) for reagents. We would like to thank the Core Services and Advanced Technologies at the Cancer Research UK Beatson Institute (C596/A17196), with particular thanks to the Beatson Advanced Imaging Resource, Biological Services Unit, Histology and Molecular Technologies. Schematic figures were created using BioRender.com.

## Author contributions

Conceived the study and designed the work plan: F.J.B., K.B. and S.W.G.T.; Experimental work: F.J.B., A.L.K., D.A., J.A., K.J.C., C.C., D.Z., E.S. and E.K; Data analysis: F.J.B., E.S., Y.F. and S.W.G.T.; Manuscript writing: F.J.B., Y.F. and S.W.G.T.

## Competing interests

The authors declare no competing interests.
