## [Peer Review File · Nature Communications]

REVIEWER COMMENTS

Reviewer #1 (Remarks to the Author):

Here the authors present an intriguing set of data to suggest that stresses which may (or may not) induce apoptosis lead to FGF2 secretion from stressed cells, resulting in FGF-mediated upregulation of pro-survival Bcl-2 family proteins (Bcl-2 and Mcl-1) in neighboring cells that protect from apoptosis. In general the data support the claims made in the paper very well and the dataset is comprehensive and convincing. However, there are some controls and additional minor experiments, as noted below, that would improve the paper before publication.

1. Given that cells do not need to die in order to induce FGF2 expression, I feel that the title is somewhat misleading and should be re-worded to 'Stress-induced FGF signaling.....'.
2. Does treatment of cells with venetoclax in the presence of FGF2 neutralizing antibodies sensitize cells to apoptosis? I know that the authors have used small molecule inhibitors of FGF2 signaling later in the paper, but there is always the chance of off-target effects with inhibitors. Use of an anti-FGF2 neutralizing antibody, or Ven-treated supernatants depleted with control antibody versus anti-FGF2 antibody and transferred onto recipient cells should clarify this.
3. Have the authors tried to generalize their findings beyond BH3-only proteins or mimetics by using other pro-apoptotic/stress signals such as chemotherapeutic drugs (taxol, cisplatin, doxorubicin, etc) to determine whether some of these also induced FGF secretion from treated cells?
4. Do we know for sure that upregulation of Bcl-2 and/or Mcl-1 were sufficient to explain the protective effects of FGF2? Could the authors knockdown expression these genes individually, or in combination, in the presence or absence of FGF2 and determine whether this is sufficient to reverse the protective effects of this growth factor against Ven?
5. Figure 3f. The lanes on the RHS of the blot (ERK1/2 CRISPR) are all underloaded as indicated by the HSP-60 loading control. This should be repeated.
6. Fig. 3g. This experiment should have S/N + Tramenitib as a control as much of the cytotoxicity observed could be due to cytotoxic effects of Tramenitib treatment. This seems to be the case as the SN Ven +Tramenitib led to more cell death than SN control.
7. It would be good to see a titration of Venetoclax versus FGF2 secretion in Figure 4. As it stands, only one dose of venetoclax has been used. It would also be useful to blot for FGF2 in Venetoclax-

treated cells over the same dose curve as it's possible that much of the FGF2 may stay cell-associated and may signal via cell-to-cell contact.

8. What S63845 is (i.e. an MCL-1 inhibitor) should be explained in the legend to Figure 1 and also re-stated throughout the text as this is not a well-known compound.

9. Page 11. There appears to be a typo here "successive repair of the skin". Did the authors mean 'successful' repair of the skin?

10. It would be useful if the authors could present the clonogenic survival data in Figure 5C as colony counts or % survival so that we can better assess the magnitude of the effects seen.

Reviewer #2 (Remarks to the Author):

The authors present an interesting manuscript identifying FGF2 as a non-cell autonomous pro-survival signal, purportedly released from apoptosing cells and acting on neighbouring cells. I think it is a valuable finding but I find some issues with their model as they currently have it, and I think the manuscript needs significant work before it is sufficiently robust to publish. My key concerns are an almost universal lack of statistical analysis of any of their data (they have a very strange style of including "Representative data from x individual experiments" at the end of each figure legend subsection). There is also an issue that their model implies apoptosis is key but their results don't show this - they can still see an effect from cells where apoptosis is blocked either by Bax/Bak CRISPR or by Caspase inhibition. This, as they allude to in the discussion, implies that there may be a non-apoptotic role for BCL-2 here, and this merits further investigation

- how is FGF2 being released?

- is FGF-2 upregulated or just secretion increased? This point definitely needs addressing.

I also struggle to see any value in their wound healing data. They don't fit at all with the study, are very poorly described, and need to be removed and replaced with much more relevant xenograft data. Further comments below relating to each figure.

The text is generally good, though they could have more in their introduction re: BH3 significance for the non-expert audience.

Fig1:

b needs quantification (as do almost all data - too many examples to highlight each one).

d no comment as to why the downregulation is so patchy - is this consistent - this is why we need to see quantification..

e needs +/- Ven in cartoon. Stats on graphs (throughout manuscript) - single lines are next to meaningless.

g very crowded - would be inclined to lose 48h data for clarity - quantify

Sup Fig1:

e (and throughout) sizes for western bands. No discussion of why PARP is so upregulated in the CRISPR cells - explain in text.

Fig2:

a include 48h in supernatant treatment in cartoon

b first two lanes not described in legend

d needs to say tBid GFP as in Sup Fig2F Quantify as significance of increase looks questionable

Sup Fig2:

a need error bars on graphs as n=3

f quantify as increase looks very marginal

Fig3:

a no info on inhibitors - need a table either here or in Sup saying what inhibitors target and what dose was used!

f Where is the (-) lane between lanes 4+5 - needs adding.

Fig 4:

d CRISPR label for FGF2.1

e why only n=2? needs another repeat and stats

g terrible westerns for FGFRs - particularly FGFR3 which does not merit inclusion. Looks like everything is down to FGFR1.

h why are there 10 lanes and 8 labels? very poor.

Sup Fig4:

Why don't they show FGFR2 WB? Can it be worse than FGFR3?

Need to show QPCR for all FGFRs given the poor quality of their WBs. I appreciate these are challenging antibodies, but not that bad.

Fig5:

I'm not an expert on bioinformatics but the data in e are poorly described - why only show thymoma - very obscure. Also they have repeated BCL2 data instead of including MCL-1

d model shouldn't have apoptosis - they don't show this.

Fig6:

All figures need to show the wound edge - this is not clear for 3dw.

c Quantitation of AZD treatment - no justification at present for comment that MCL-1 is regulated.

No mention that vast majority of excisional wound healing comes from connective tissue contraction - they need to measure the area of the wound epidermis (although I would suggest wh data don't add anything.)

Sup Fig6:

Again, wound edges are not clear. MCM2 and Ki67 seems like tautology

c actin and Ki67 are labelled incorrectly throughout.

This last figure needs replacing with xenograft data with relevant cancer cell lines into which inhibitors have been injected, or better still using CRISPR cells which can't produce FGF2 or which lack FGFR1.

Reviewer #1:

“Here the authors present an intriguing set of data to suggest that stresses which may (or may not) induce apoptosis lead to FGF2 secretion from stressed cells, resulting in FGF-mediated upregulation of pro-survival Bcl-2 family proteins (Bcl-2 and Mcl-1) in neighboring cells that protect from apoptosis. In general the data support the claims made in the paper very well and the dataset is comprehensive and convincing. However, there are some controls and additional minor experiments, as noted below, that would improve the paper before publication.”

Response: Thank you for the critical evaluation of our manuscript, we appreciate the positive feedback. Through addressing the points raised, we consider our manuscript much improved.

“1. Given that cells do not need to die in order to induce FGF2 expression, I feel that the title is somewhat misleading and should be re-worded to 'Stress-induced FGF signaling.....'.”

Response: We agree with the reviewer's comment and have re-worded the title to *“Apoptotic stress-induced FGF signaling promotes non-cell autonomous resistance to cell death”*.

“2. Does treatment of cells with venetoclax in the presence of FGF2 neutralizing antibodies sensitize cells to apoptosis? I know that the authors have used small molecule inhibitors of FGF2 signaling later in the paper, but there is always the chance of off-target effects with inhibitors. Use of an anti-FGF2 neutralizing antibody, or Ven-treated supernatants depleted with control antibody versus anti-FGF2 antibody and transferred onto recipient cells should clarify this.”

Response: To further investigate the finding that FGF2 is responsible for paracrine resistance, in new experiments (**Figure 6d**) we find that transfer of supernatant from FGF2 deficient cells treated with venetoclax significantly inhibits protection against cell death. These data further support the hypothesis that FGF2 is responsible for promoting paracrine survival in response to stress helping to exclude potential off-target effects of inhibitors.

“3. Have the authors tried to generalize their findings beyond BH3-only proteins or mimetics by using other pro-apoptotic/stress signals such as chemotherapeutic drugs (taxol, cisplatin, doxorubicin, etc) to determine whether some of these also induced FGF secretion from treated cells?”

Response: Thank you for this important suggestion, in new experiments we find upregulation of MCL1 and BCL-2 following supernatant transfer from cells treated with etoposide, doxorubicin and paclitaxel (**Figure 3h**). These data argue that the described pathway of paracrine resistance is also engaged in response to conventional chemotherapies, expanding the generality of our findings.

“4. Do we know for sure that upregulation of Bcl-2 and/or Mcl-1 were sufficient to explain the protective effects of FGF2? Could the authors knockdown expression these genes individually, or in combination, in the presence or absence of FGF2 and determine whether this is sufficient to reverse the protective effects of this growth factor against Ven?”

Response: Our data in **Figure 6a/b** shows that inhibition of MCL1 and BCL-2 upregulation using inhibitors of MEK1/2 and FGF receptors re-sensitises cells. These inhibitors efficiently prevent upregulation of MCL1 and BCL-2 (**Figure 3b, 3d, 4f, S4b**) In new experiments (**Figure 6d**) we find that transfer of supernatant from FGF2 CRISPR cells, which is incapable of upregulating MCL1 and BCL-2 (**Figure 4d**) also fails to promote resistance. Collectively these data support a tight correlation between FGF2 mediated apoptotic protection and BCL-2/MCL1 upregulation. Supporting functional relevance, we find that addition of the MCL1 inhibitor (S63845) (**Figures 6a, b**) or increasing the dose of venetoclax and S63845 (**Figures 6a, b**) overcomes FGF-2 mediated resistance, strongly arguing for the functional importance of MCL1 and BCL-2 upregulation in FGF2 mediated resistance.

“5. Figure 3f. The lanes on the RHS of the blot (ERK1/2 CRISPR) are all underloaded as indicated by the HSP-60 loading control. This should be repeated.”

Response: As requested, this blot has been repeated with equal loading (**Figure 3f**).

“6. Fig. 3g. This experiment should have S/N + Trametinib as a control as much of the cytotoxicity observed could be due to cytotoxic effects of Trametinib treatment. This seems to be the case as the SN Ven +Trametinib led to more cell death than SN control.”

Response: In new experiments (**Figures 6a, 6b, S5a and S5b**) we provide experimental data from cells treated with trametinib and FGFR inhibitors following direct treatment and in the supernatant transfer setting. These experiments showed that trametinib or FGFR inhibitor addition does not have a major effect on cell death.

“7. It would be good to see a titration of Venetoclax versus FGF2 secretion in Figure 4. As it stands, only one dose of venetoclax has been used. It would also be useful to blot for FGF2 in Venetoclax-treated cells over the same dose curve as it's possible that much of the FGF2 may stay cell-associated and may signal via cell-to-cell contact.”

Response: In a new experiment, we have repeated the ELISA in **Figure 4b** with one showing FGF2 levels in the supernatant also following treatment with a lower dose of venetoclax. Additionally, by western blot, we now show in **Figure 5b** that cellular levels of FGF2 decrease following venetoclax treatment, arguing that release of FGF2 from cells is likely the key step affected by venetoclax treatment.

“8. What S63845 is (i.e. an MCL-1 inhibitor) should be explained in the legend to Figure 1 and also re-stated throughout the text as this is not a well-known compound.”

Response: We have added more information about S63845 throughout the text and the legend.

“9. Page 11. There appears to be a typo here "successive repair of the skin". Did the authors mean 'successful' repair of the skin?”

Response: Thank you for highlighting this mistake, we have corrected it.

"10. It would be useful if the authors could present the clonogenic survival data in Figure 5C as colony counts or % survival so that we can better assess the magnitude of the effects seen."

Response: We have now quantified the data from clonogenic survival assays and present this data in **Figure 6c**.

Reviewer #2

"The authors present an interesting manuscript identifying FGF2 as a non-cell autonomous pro-survival signal, purportedly released from apoptosing cells and acting on neighbouring cells. I think it is a valuable finding but I find some issues with their model as they currently have it, and I think the manuscript needs significant work before it is sufficiently robust to publish. My key concerns are an almost universal lack of statistical analysis of any of their data (they have a very strange style of including "Representative data from x individual experiments" at the end of each figure legend subsection). There is also an issue that their model implies apoptosis is key but their results don't show this - they can still see an effect from cells where apoptosis is blocked either by Bax/Bak CRISPR or by Caspase inhibition. This, as they allude to in the discussion, implies that there may be a non-apoptotic role for BCL-2 here, and this merits further investigation"

Response: We thank the reviewer for their constructive and positive review. By addressing the Reviewer's comments we consider the manuscript much improved.

"- how is FGF2 being released?

- is FGF-2 upregulated or just secretion increased? This point definitely needs addressing."

Response: We have carried out new experiments to address these points (described in **Figure 5**). We now provide evidence that venetoclax treatment has only a small effect upon transcription of FGF2 (**Figure 5a**), but protein levels decrease slightly. Secondly we find that blocking of transcription in producer cells had no effect upon ventoclax induced MCL-1 upregulation in a paracrine manner (**Figure 5c**). These data

demonstrate that the effect of BCL-2 inhibition upon FGF2 induced MCL-1 expression is independent of synthesis, implying that BCL-2 inhibition promotes FGF2 release (discussed on page 16). Following intense research efforts by many labs, the mechanisms regulating FGF2 intracellular trafficking and release remain poorly understood. In turn, this makes it very challenging to define how BCL-2 inhibition affects this ill-defined process. Nevertheless, as we discuss, further understanding of how BCL-2 regulates FGF2 processing or release may provide fresh insight into physiological regulation of these processes.

“I also struggle to see any value in their wound healing data. They don't fit at all with the study, are very poorly described, and need to be removed and replaced with much more relevant xenograft data. Further comments below relating to each figure.”

Response: We thank the reviewer for raising this point, we apologise for not fully describing the rationale for pursuing the wound model in the initial manuscript (this is now revised). We reasoned, as with most oncogenic processes, there must be physiological role(s) for apoptotic stress induced FGF2 relayed anti-apoptotic BCL-2 expression. A possible function in wound healing struck as a particular relevant context, especially given the oft drawn analogy between tumourigenesis as a wound that does not heal. In new experiments (described below) we also carried out a xenograft experiment as suggested by the reviewer.

“The text is generally good, though they could have more in their introduction re: BH3 significance for the non-expert audience.”

Response: In the revised manuscript, we have expanded the section on BH3 mimetics in the introduction (**lines 53 - 58**).

“Fig1:

b needs quantification (as do almost all data - too many examples to highlight each one).”

Response: We agree with the reviewer's comment and have now added quantification and statistical analysis throughout the manuscript.

“d no comment as to why the downregulation is so patchy - is this consistent - this is why we need to see quantification..”

Response: The experiment has been repeated multiple times, demonstrating a consistent downregulation (**Figure 1d**).

“1e needs +/- Ven in cartoon. Stats on graphs (throughout manuscript) - single lines are next to meaningless.”

Response: We have added \pm Ven to the cartoon and now show combined data from multiple independent repeats for the majority of experiments described in the manuscript.

“1g very crowded - would be inclined to lose 48h data for clarity – quantify”

Response: For clarity, as the reviewer suggests, we have cropped the blot to only show the 6 h and 24 h timepoints. We also now present a more comprehensive blot of the corresponding supernatant transfer experiment following tBID transfection (**Figure 2d**).

“Sup Fig1e (and throughout): sizes for western bands”.

Response: We apologize for this oversight, we have now added molecular weight markers for the western blots throughout.

“No discussion of why PARP is so upregulated in the CRISPR cells - explain in text.”

Response: We are unclear as to why full length PARP1 is decreased in control cells in this blot. As we note, multiple assays show that BAX and BAK deleted cells are resistant to apoptosis (cleavage of several apoptotic caspases and survival assay), in accordance with their well described essential role in apoptosis.

“Fig2:

2a include 48h in supernatant treatment in cartoon”

Response: This is now added to the schematic.

“2b first two lanes not described in legend”

Response: This is now added to the legend.

“2d needs to say tBid GFP as in Sup Fig2F Quantify as significance of increase looks questionable”

Response: We have added the caption for tBID GFP and quantified this experiment.

“Sup Fig 2: a need error bars on graphs as n=3”

Response: We have added error bars for this experiment.

“Sup Fig 2 f quantify as increase looks very marginal”

Response: This experiment has now been quantified.

“Fig 3: a no info on inhibitors - need a table either here or in Sup saying what inhibitors target and what dose was used!”

Response: To aid the reader we have now added this information to the Materials and Methods section (line 615 - 627).

“3f Where is the (-) lane between lanes 4+5 - needs adding.”

Response: We have repeated the experiment including the additional (-) sample (**Figure 3f**).

“Fig 4d: CRISPR label for FGF2.1”

Response: We have changed the labelling accordingly.

“4e why only n=2? needs another repeat and stats”

Response: This has been repeated, statistical analysis demonstrates a significant difference.

“4g terrible westerns for FGFRs - particularly FGFR3 which does not merit inclusion. Looks like everything is down to FGFR1.”

Response: Due to consistent lack of reliable detection of FGFR3 and FGFR4 by western blot we have now included qRT PCR data for FGFR3 and FGFR4.

“4h why are there 10 lanes and 8 labels? very poor.”

Response: Upon submission we noted this oversight - our apologies - we have replaced the incorrect loading control with the correct one.

“Sup Fig4: Why don't they show FGFR2 WB? Can it be worse than FGFR3?”

Response: We repeatedly failed to detect FGFR2 by western blot, this led us to query RNA-seq data, where FGFR2 mRNA was not detectable (**Figure S4d**). We have therefore removed these experiments and adjusted the text accordingly to note this (**line 246**).

“Need to show QPCR for all FGFRs given the poor quality of their WBs. I appreciate these are challenging antibodies, but not that bad.”

Response: Thank you for this suggestion, we have now included qRT-PCR for FGFR3 and FGFR4 siRNA experiments (**Figure S4e, h**).

“Fig5:

I'm not an expert on bioinformatics but the data in e are poorly described - why only show thymoma - very obscure. Also they have repeated BCL2 data instead of including MCL-1"

Response: In the revised manuscript we have better described our bioinformatic analysis, discussing that we analysed 25 tumour types for potential correlation between an FGF activation signature and anti-apoptotic BCL2 expression. Thymoma was one example where we found a correlation between FGFR activation and BCL2-MCL1 levels, and where patients with higher levels of BCL2 or MCL-1 and FGF activation had a poorer prognosis. Further investigation revealed additional tumour types (Chromophobe renal cell carcinoma and Uterine corpus endometrial carcinoma) where this was the case, these are now described in **Supplemental Figure 6a and b** in the revised manuscript. Upon submission, we noted the MCL-1 data was not included, we apologise for the oversight - this has been sorted in the revised version.

"5d model shouldn't have apoptosis - they don't show this."

Response: We have altered the figure and modified text (**line 353 - 354**) accordingly

"Fig6:

All figures need to show the wound edge - this is not clear for 3dw."

Response: Wound edges are now added to all figures

"6c Quantitation of AZD treatment - no justification at present for comment that MCL-1 is regulated."

Response: We have now quantified MCL-1 expression finding that it is downregulated following AZD4547 and trametinib treatment (**Figure 7c**).

"No mention that vast majority of excisional wound healing comes from connective tissue contraction - they need to measure the area of the wound epidermis (although I would suggest wh data don't add anything.)"

Response: We now have added quantification of the leading edge length (**Supplemental Figures 7b, c**) and adjusted the text to discuss the importance of tissue contraction as the reviewer notes.

“Sup Fig 6: Again, wound edges are not clear. MCM2 and Ki67 seems like tautology c actin and Ki67 are labelled incorrectly throughout.”

Response: We apologise for the labelling error (this has been corrected) and have added indications for the wound edge. Analysis of both MCM2 and Ki67 provides independent makers to validate effects on proliferation.

“This last figure needs replacing with xenograft data with relevant cancer cell lines into which inhibitors have been injected, or better still using CRISPR cells which can't produce FGF2 or which lack FGFR1.”

Response:

We thank the referee for this suggestion, and performed xenograft experiments using mitoprimes control and FGF2 deficient HeLa cells. While *in vitro* we did not observe any issues in proliferation of FGF2 deficient cells (**Reviewer Figure 1**), *in vivo* these cells failed to thrive after engraftment and regressed over time (**Reviewer Figure 2**). These data argue that pro-survival signalling mediated by FGF2 may be required for tumour growth. In an alternative approach, we investigated the capacity of trametinib to enhance BH3-mimetic treatment, unfortunately, for unclear reasons this led to rapid weight loss in the experimental cohort necessitating culling.

Reviewer Figure 1:

Growth of HeLa tBID-2A-BCL-2 EMPTY^{CRISPR} or FGF2.2^{CRISPR} cells was monitored via Incucyte live cell imaging.

Reviewer Figure 2:

Two million HeLa tBID-2A-BCL-2 EMPTY^{CRISPR} or FGF2.2^{CRISPR} cells were injected subcutaneously into CD1 nude mice and tumor growth was monitored over time.

REVIEWER COMMENTS

Reviewer #1 (Remarks to the Author):

The authors are to be congratulated on a thorough response to my previous comments, which have addressed all of the issues raised.

One minor comment regarding the title, which should have a hyphen inserted between 'stress induced', as in 'Apoptotic stress-induced FGF signaling....'

Reviewer #2 (Remarks to the Author):

The authors have made considerable improvements to the manuscript following significant further work. They have corrected the multiple errors in the first submission, following referee comments that addressed several issues that should have been addressed prior to submission. They persist in submitting wound healing data that are poorly described. For example, they state that wound edges are included in all figures - this is patently wrong. Where is the wound edge in 7a 0 and 3d PWI? The hyperplastic migrating epidermis is clear at d5 but they still crop the tip of it on the right hand side, In b they need to show lower mag as well if they are claiming the line is where the initial wound was - would be useful to see H&E staining. For the AZD treatment this wound site is wrong, since there is a clear hair follicle to the right of the excisional wound site.

Reviewer #1

"The authors are to be congratulated on a thorough response to my previous comments, which have addressed all of the issues raised.

One minor comment regarding the title, which should have a hyphen inserted between 'stress induced', as in 'Apoptotic stress-induced FGF signaling....'"

Response: This is now changed in the revised version.

Reviewer #2

"The authors have made considerable improvements to the manuscript following significant further work. They have corrected the multiple errors in the first submission, following referee comments that addressed several issues that should have been addressed prior to submission. They persist in submitting wound healing data that are poorly described. For example, they state that wound edges are included in all figures - this is patently wrong. Where is the wound edge in 7a 0 and 3d PWI? The hyperplastic migrating epidermis is clear at d5 but they still crop the tip of it on the right hand side, In b they need to show lower mag as well if they are claiming the line is where the initial wound was - would be useful to see H&E staining. For the AZD treatment this wound site is wrong, since there is a clear hair follicle to the right of the excisional wound site."

Response: We thank the reviewer for highlighting these points and apologise for the lack of clarity. Taking the Reviewer's suggestions into account we have repeated staining of tissue sections, using serial sections and H&E staining, to better highlight on the H&E stain where the wound infliction was made (denoted by broken-line), and denoting wound site/unwounded skin. These images replace the previous images shown on Figure 7. In the previous version of Figure 7, 7a day 0 PWI, was control (unwounded skin), this has now been annotated "wild-type dorsal skin", to avoid confusion that this was wounded skin.

REVIEWERS' COMMENTS

Reviewer #2 (Remarks to the Author):

The authors have addressed my concerns with the wound healing data - though I don't think they really address why there is the 40% difference in healing at day 1? Looking at the methods it may well be a contraction defect (due to the pre-treatment with inhibitors), since this is a major driver for murine excisional wound healing. So I feel they are missing a trick in not addressing a potential role on fibroblasts - but the rest of their manuscript is a well constructed and convincing narrative. It would have been nice to see alpha-smooth muscle actin staining of wounds to confirm this, but I appreciate that they are not particularly focussed on performing a comprehensive wound healing study.

Reviewer #2

The authors have addressed my concerns with the wound healing data - though I don't think they really address why there is the 40% difference in healing at day 1? Looking at the methods it may well be a contraction defect (due to the pre-treatment with inhibitors), since this is a major driver for murine excisional wound healing. So I feel they are missing a trick in not addressing a potential role on fibroblasts - but the rest of their manuscript is a well constructed and convincing narrative. It would have been nice to see alpha-smooth muscle actin staining of wounds to confirm this, but I appreciate that they are not particularly focussed on performing a comprehensive wound healing study.

Response: We thank the reviewer for their comments.